# Sites and mechanisms of action of colokinetics at dopamine, ghrelin and serotonin receptors in the rodent lumbosacral defecation centre

Mitchell T. Ringuet[1] , Ada Koo[1] , Sebastian G. B. Furness[2,4] , Stuart J. McDougall[3]
and John B. Furness[1,3]

[1] *Department of Anatomy & Physiology, University of Melbourne, Melbourne, VIC, Australia*
[2] *School of Biomedical Sciences, University of Queensland, Brisbane, QLD, Australia*
[3] *Florey Institute of Neuroscience and Mental Health, University of Melbourne, Melbourne, VIC, Australia*
[4] *Monash Institute of Pharmaceutical Sciences, Melbourne, VIC, Australia*

Handling Editors: Peying Fong & Bernard Drumm

The peer review history is available in the Supporting Information section of this article (https://doi.org/10.1113/JP285217#support-information-section).

The Journal of Physiology

**Abstract** Agonists of dopamine D2 receptors (D2R), 5-hydroxytryptamine (5-HT, serotonin) receptors (5-HTR) and ghrelin receptors (GHSR) activate neurons in the lumbosacral defecation centre, and act as 'colokinetics', leading to increased propulsive colonic motility, *in vivo*. In the present study, we investigated which neurons in the lumbosacral defecation centre express the receptors and whether dopamine, serotonin and ghrelin receptor agonists act on the same lumbosacral preganglionic neurons (PGNs). We used whole cell electrophysiology to record responses from

**Mitchell Ringuet** is a recent PhD graduate from the Department of Anatomy and Physiology at the University of Melbourne, working under Professor John Furness. His doctoral project has focused on the interaction between G protein-coupled receptors, mainly GHSR and D2R. He completed his Master of Science in 2014 at the University of Melbourne working on inflammatory pain pathways and went on to work in the fields of synaptic neurobiology and digestive physiology before commencing his PhD.

neurons in the lumbosacral defecation centre, following colokinetic application, and investigated their expression profiles and the chemistries of their neural inputs. Fluorescence *in situ* hybridisation revealed *Drd2*, *Ghsr* and *Htr2C* transcripts were colocalised in lumbosacral PGNs of mice, and immunohistochemistry showed that these neurons have closely associated tyrosine hydroxylase and 5-HT boutons. Previous studies showed that they do not receive ghrelin inputs. Whole cell electrophysiology in adult mice spinal cord revealed that dopamine, serotonin, $\alpha$-methylserotonin and capromorelin each caused inward, excitatory currents in overlapping populations of lumbosacral PGNs. Furthermore, dopamine caused increased frequency of both IPSCs and EPSCs in a cohort of D2R neurons. Tetrodotoxin blocked the IPSCs and EPSCs, revealing a post-synaptic excitatory action of dopamine. In lumbosacral PGNs of postnatal day 7–14 rats, only dopamine's postsynaptic effects were observed. Furthermore, inward, excitatory currents evoked by dopamine were reduced by the GHSR antagonist, YIL781. We conclude that lumbosacral PGNs are the site where the action of endogenous ligands of D2R and 5-HT2R converge, and that GHSR act as a cis-modulator of D2R expressed by the same neurons.

(Received 4 July 2023; accepted after revision 13 September 2023; first published online 28 September 2023)

**Corresponding author** M. T. Ringuet: Department of Anatomy & Physiology, University of Melbourne, Melbourne, VIC, 3010, Australia.    Email: mringuet@student.unimelb.edu.au

**Abstract figure legend** Dopamine, serotonin and ghrelin regulate neuronal excitability in the rodent lumbosacral defecation centre. *In vivo*, agonists of dopamine D2 receptors (D2R), 5-hydroxytryptamine type 2 (5-HT2) and ghrelin receptors (GHSR) lead to increased propulsive colonic motility. We found that a subset of preganglionic neurons (PGNs) express D2R, 5-HT2 and GHSR receptors and that dopamine, serotonin and ghrelin receptor agonists each increased neuronal excitability. We discovered that a role of the ghrelin receptor is to reverse the effect of dopamine at D2R from inhibition to excitation. Created with Biorender.com.

### Key points

- Dopamine, 5-hydroxytryptamine (5-HT, serotonin) and ghrelin (GHSR) receptor agonists increase colorectal motility and have been postulated to act at receptors on parasympathetic pre-ganglionic neurons (PGNs) in the lumbosacral spinal cord.
- We aimed to determine which neurons in the lumbosacral spinal cord express dopamine, serotonin and GHSR receptors, their neural inputs, and whether agonists at these receptors excite them.
- We show that dopamine, serotonin and ghrelin receptor transcripts are contained in the same PGNs and that these neurons have closely associated tyrosine hydroxylase and serotonin boutons.
- Whole cell electrophysiology revealed that dopamine, serotonin and GHSR receptor agonists induce an inward excitatory current in overlapping populations of lumbosacral PGNs. Dopamine-induced excitation was reversed by GHSR antagonism.
- The present study demonstrates that lumbosacral PGNs are the site at which actions of endogenous ligands of dopamine D2 receptors and 5-HT type 2 receptors converge. Ghrelin receptors are functional, but their role appears to be as modulators of dopamine effects at D2 receptors.

## Introduction

Autonomic control centres in the spinal cord, notably the intermediolateral (IML) cell groups, receive both serotoninergic and dopaminergic inputs from the lower brain stem (Anderson et al., 2009; Hinrichs & Llewellyn-Smith, 2009; Sawamura et al., 2023). In addition, a high proportion of neurons in the IML express the ghrelin receptor (GHSR) (Ferens, Yin, Bron et al., 2010), and centrally penetrant ghrelin receptor agonists activate autonomic pathways, apparently through excitation of preganglionic neurons (PGNs) in the IML. The pathways activated include vasoconstrictor (Ferens, Yin, Bron et al., 2010), micturition (Ferens, Yin, Ohashi-Doi et al., 2010) and defecation circuits (Naitou et al., 2015; Shimizu et al., 2006). This apparent role of the ghrelin receptor in spinal autonomic centres stands in contrast to the absence of the natural agonist, ghrelin,

in the spinal cord (Furness et al., 2011; Pustovit et al., 2017). Recent investigations suggest that, in the absence of an endogenous ligand, a role of GHSR in the central nervous system (CNS) might be to modulate the actions of neurotransmitters, including the effects of dopamine at dopamine D2 receptor (D2R) (Hedegaard & Holst, 2020; Kern et al., 2012, 2014; Price et al., 2021; Ringuet et al., 2021). In the case of D2R, the presence of GHSR is proposed to reverse its normally inhibitory effect on neurons to excitation (Kern et al., 2014; Ringuet et al., 2021). In the present study, we have used neurons in the lumbosacral spinal cord to investigate the convergence of dopamine and serotonin inputs at autonomic nuclei, as well as the possible role of GHSR in determining that the effects of dopamine are excitatory in these nuclei.

Lumbosacral PGNs, local interneurons (together forming part of the lumbosacral defecation centre) and motor neurons of Onuf's nucleus in the lumbosacral spinal cord (L6–S1) work in a co-ordinated fashion to control colorectal function (Callaghan et al., 2018). When administered to the lumbosacral defecation centre in rats, the biogenic amines, dopamine, through D2R (Furness et al., 2021; Naitou et al., 2016) and 5-hydroxytryptamine (5-HT, serotonin) through both metabotropic 5-HT type 2 (5-HT2) and ionotropic 5-HT type 3 (5-HT3) receptors (Nakamori, Naitou, Sano et al., 2018) stimulate propulsive colorectal motility. In both cases, this response is prevented by cutting the pelvic nerves between the lumbosacral spinal cord and the colorectum, implying that the agonists do indeed exert their effects through the lumbosacral defecation centre. In the case of dopamine and 'D2R preferring' agonists, whole cell recordings in rats (1–3 weeks old) showed that they excited PGNs in the lumbosacral defecation centre (Naitou et al., 2016). This is a surprising result because dopamine acting at D2R is generally inhibitory (Neve et al., 2004). However, it has been shown that inhibitory effects at D2R may be reversed to excitation in neurons also expressing the GHSR (Kern et al., 2012; Ringuet et al., 2021). GHSR is expressed in many of the same lumbosacral PGNs as D2R in rats and functional interaction of both receptors has been shown *in vivo*, in which GHSR antagonism blocks dopamine induced increases in colonic motility (Furness et al., 2021). The site of this antagonism is unknown. In the present study, we have utilised spinal cord from D2R reporter mice to investigate whether lumbosacral PGNs express dopamine (*Drd2*), serotonin (*Ht2rc*) and ghrelin (*Ghsr*) G protein-coupled receptor (GPCR) transcript; whether D2R neurons receive aminergic inputs; and whether agonists, dopamine, serotonin, $\alpha$-methylserotonin ($\alpha$-MS) and capromorelin (a small molecule GHSR agonist) excite the same neurons. Previous key experiments that suggest that these amines and ghrelin could act at a common site were obtained in rats (Naitou et al., 2016; Nakamori et al., 2019; Shimizu

et al., 2006). We have therefore also used lumbosacral spinal cord slices from neonatal [postnatal day (P)7–14] rats to investigate whether dopamine is excitatory at lumbosacral PGNs and whether, as suggested by *in vivo* rat experiments, excitatory effects of dopamine are dependent on ghrelin receptors (Furness et al., 2021).

## Methods

Experiments were conducted in accordance with the National Health and Medical Research Council guidelines for the care and use of animals and with approval from the Florey Institute of Neuroscience and Mental Health Animal Ethics Committee (approval FINMH-20-024).

### Animals

For histology and whole cell recordings, 36 adult (female and male) drd2-tdTomato mice (D2R; aged 8−20 weeks) and three 8-week-old C57BL/6JArc mice (two female and one male) were used. D2R reporter mice were generated by crossing heterozygous Drd2-cre MMRRC: (B6.FVB(Cg)-Tg (Drd2-cre) ER44Gsat/Mmucd) with homozygous tdTomato (Ai14(RCL-tdT)-D) animals to yield heterozygous: heterozygous drd2-tdTomato (D2R) reporter mice. In total, 26 (male and female) Sprague Dawley rats (P4–11) were injected I.P. with 10 $\mu$L (20 $\mu$g $\mu$L$^{-1}$ dissolved in dimethyl sulfoxide) of the retrograde tracer, Fast DiI (1,1′-dilinoleyl-3,3,3′,3′-tetramethylindocarbocyanine, 4-chlorobenzenesulphonate, D7756; Invitrogen, Waltham, MA, USA) to back label parasympathetic PGNs in the lumbosacral spinal cord (Anderson & Edwards, 1994). After 3−7 days, Sprague–Dawley neonatal (P7–14) rats were killed for slice preparation (see below) and subsequent whole cell electrophysiology.

### Immunohistochemistry (localisation and synaptic input)

Drd2-tdTomato (D2R) mice (aged 8−20 weeks) were anaesthetised by I.P. injection of ketamine hydrochloride (100 mg kg$^{-1}$) plus xylazine hydrochloride (20 mg kg$^{-1}$) and transcardially perfused with phosphate-buffered saline (PBS), followed by 4% paraformaldehyde (PFA) at 4°C. Lumbosacral spinal cords (L6–S1) were dissected, post-fixed overnight (O/N), washed with PBS (3 × 10 min) and frozen in optimal cutting temperature compound (OCT) for sectioning. Coronal sections (15 $\mu$m) were cut, air-dried for 1 h on SuperFrostPlus slides (Thermo Fisher, Scoresby, VIC, Australia) and incubated with 10% Normal Horse Serum (#31874; Invitrogen) at room temperature (RT) for 30 min. Sections were then incubated with primary antisera

at 4°C O/N. The following day, the tissue was washed with PBS (3 × 10 min) and incubated in secondary antisera for 2 h at RT. For staining nuclei, preparations were washed once with PBS, twice with $dH_2O$, followed by a 5 min of incubation in Hoechst 33258 solution (Bisbenzimide–Blue, diluted to 10 $\mu g$ mL$^{-1}$ in $dH_2O$). Excess Hoechst was removed with $dH_2O$ (3 × 5 min) before mounting coverslips with Dako non-fluorescent mounting medium (Dako, Carpinteria, CA, USA).

**List of primary and secondary antisera used for immunohistochemistry.**

| Primary antisera | Source | Dilution | Secondary antisera | Source | Dilution |
|---|---|---|---|---|---|
| Goat anti-ChAT | Chemicon; AB144P; Lot #23 061 013 (Temecula, CA, USA) | 1:200 | Dk anti-goat 647 plus | Invitrogen; A32849; Lot #VB295511 | 1:500 |
| Rabbit anti-TH | Thibault; A52-512; Institute Jacques Bay (Gift of Jean Thibault Laboratorie de Biochimie Cellularie, College de France, Paris, France) | 1:1,000 | Dk anti-rabbit 488 | Invitrogen; A21206; Lot #1 874 771 | 1:800 |
| Goat anti-5-HT | Incstar; 20 079; Lot #1 906 001 (Stillwater, MN, USA) | 1:10 000 | Dk anti-goat 647 plus | Invitrogen; A32849; Lot #VB295511 | 1:500 |

### RNAscope

RNAscope assays were performed to identify lumbosacral PGNs containing *Drd2*, *Ghsr* and *ChAT* mRNA transcripts. This was carried out in accordance with the user manual for Fresh Frozen Tissue using the RNAscope Multiplex Fluorescent Reagent Kit (Cat. No. 320 850; Advanced Cell Diagnostics, Hayward, CA, USA). Briefly, 8-week-old C57BL/6JArc mice (two female and one male) were anaesthetised using isoflurane (5%) and killed by severing their carotid arteries and cervical spinal cord. The lumbosacral spinal cord (L6–S1) was quickly removed (∼5 min) and frozen on dry ice. Tissue was frozen in OCT using isopentane that was cooled with liquid nitrogen and coronal sections (12 $\mu m$) were cut and thaw mounted directly on SuperFrostPlus slides (Thermo Fisher) and stored at −20°C. Slides were removed from the freezer and tissue sections were fixed by immersion in 4% PFA for 15 min at 4°C. They were then rinsed twice in PBS and dehydrated in 50% and 70% ethanol (EtOH) solution for 5 min each, followed by dehydration in 100% EtOH twice for 5 min at RT. Slides were then transferred to 100% EtOH and kept at −20°C O/N. The following day, slides were air dried at RT for 10 min and incubated with

protease solution (Cat No. 322 336; RNAscope Protease IV solution) for 30 min at RT. Excess solution was removed, and slides were washed twice in PBS. Target probes for *Mus musculus dopamine receptor d2* (*Drd2*) (Cat No. 406 501-C3, NM_01 0077.2, target region: 69–1175), *Mus musculus growth hormone secretagogue receptor* (*Ghsr*) (Cat No. 426 141, NM_177 330.4, target region: 483–1385) and *Mus musculus choline acetyltransferase* (*Chat*) (Cat No. 408 731-C2, NM_0 09891.2, target region: 1090–1952) were applied to the sections and incubated at 40°C for 2 h in the HybEZ Hybridization System (Advanced Cell Diagnostics). Slides were then washed 2 × 2 min in wash buffer solution before amplification steps. Sections were incubated with amplifier probes by applying AMP1 (40°C for 30 min), AMP2 (40°C for 15 min) and AMP3 (40°C for 30 min), washing slides in wash buffer solution (2 × 2 min) between addition of amplifier probes. Sections were then incubated with the AMP4 Alt C-FL fluorescence labelled probe to detect target probes (Alexa Fluor 488 – *Drd2*, ATTO 550 – *Ghsr*, ATTO 647 – *Chat*). Excess amplifier probe was removed, and slides were washed 2 × 2 min in wash buffer solution. Slides were then incubated in 4′,6-diamidino-2-phenylindole (DAPI) for 30 s to stain nuclei and mounted using Prolong® Gold Antifade Reagent (P36930; Life Technologies, Carlsbad, CA, USA). Negative control sections were incubated with RNAscope 3-Plex Negative Control Probe (Cat No. 320 871) and positive control sections were incubated with RNAscope 3-plex Positive Control Probe – Mm (Cat No. 320 881).

### High-resolution RNA fluorescence *in situ* hybridisiation (HCR-RNA FISH)

To identify *Htr2c* in D2R neurons, 12-week-old drd2-tdTomato (D2R) mice were anaesthetised by I.P. injection of ketamine hydrochloride (100 mg kg$^{-1}$) plus xylazine hydrochloride (20 mg kg$^{-1}$) and transcardially perfused with PBS, followed by 4% PFA at 4°C. Lumbosacral spinal cord (L6–S1) was dissected, post-fixed O/N, washed with PBS (3 × 10 min) and snap frozen in OCT using isopentane that was cooled with liquid nitrogen. Coronal sections (15 $\mu m$) were cut, and thaw mounted directly on Superfrost Plus slides (Thermo Fisher) and stored at −80°C. HCR RNA-FISH was performed in accordance with the user manual (Molecular Instruments, Los Angeles, CA, USA). Slides were taken from −80°C and brought to RT. Sections were treated with 10 $\mu g$ mL$^{-1}$ proteinase K (25530-015; Invitrogen) for 10 min at 37°C. Slides were washed twice in PBS, followed by 100 $\mu L$ of pre-hybridisation buffer to sections for 10 min at 37°C. Pre-hybridisation buffer was drained and 100 $\mu L$ of *Mus musculus Ht2rc* (NM_0 08312.4; 1.6 pmol) was added to the section

with parafilm placed on top O/N at 37°C to prevent evaporation. The following day, slides were immersed in wash buffer at 37°C. Slides were then incubated in a series of pre-warmed wash buffer/saline-sodium citrate and tween (SSCT; 25% 20 × SSC, 0.1% Tween-20 in DNAse/RNAse-free dH$_2$O) combinations at 37°C: 75% probe wash buffer/25% 5 × SSCT for 15 min; 50% probe wash buffer/50% 5 × SSCT for 15 min; 25% probe wash buffer/75% 5 × SSCT for 15 min; and 100% 5 × SSCT for 15 min. Slides were then immersed in 5 × SSCT for 5 min at RT. For amplification, 100 $\mu$L amplification buffer was added to sections for 30 min at RT. Hairpin solution was prepared by adding snap-cooled h1 and h2 hairpins to 100 $\mu$L of amplification buffer at RT (for snap cooling method refer to user manual). Pre-amplification solution was removed and 100 $\mu$L of hairpin solution was added and incubated O/N with parafilm in a dark humidified box at RT. The following day excess hairpins were removed by incubating slides in 5 × SSCT for 2 × 30 min, followed by 1 × 5 min at RT. Slides were then incubated in DAPI to stain nuclei, washed and mounted using DAPI.

## Solutions and slice preparation

D2R reporter mouse and neonatal (P7–14) rat spinal cord slices were prepared in the same way. Animals were anaesthetised I.P. with ketamine hydrochloride (100 mg kg$^{-1}$) followed by decapitation and exsanguination. A dorsal laminectomy was performed on ice to remove the lumbosacral spinal cord (L6–S1). Dura mater and dorsal/ventral roots were trimmed in a Sylgard dish before the spinal cord was glued to a 4% agar block, ventral side down, and then to the specimen disc, with cyanoacrylate. Coronal sections (300 $\mu$m) were cut on a Vibratome VT1200S (Leica, Wetzlar, Germany) in ice-cold (2–4°C) sucrose-substituted artificial cerebrospinal fluid pre-oxygenated with carbogen (saCSF; 350 mOsm kg$^{-1}$), containing (in mM); 250 sucrose, 25 NaHCO$_3$, 10 D-glucose, 2.5 KCl, 1 NaH$_2$PO$_4$, 1 MgCl$_2$ and 2 CaCl$_2$. Slices were then transferred into a carbogenated NMDG-Hepes recovery solution (NMDG-aCSF; 300 mOsm kg$^{-1}$) for 2−15 min at 32°C, containing (in mM); 93 NMDG, 30 NaHCO$_3$, 20 Hepes, 25 D-glucose, 2.5 KCl, 1.2 NaH$_2$PO$_4$, 2 thiourea, 3 Na pyruvate, 10 MgSO$_4$, 1 CaCl$_2$ and 5 Na ascorbate (Ting et al., 2018). Slices were then placed in recording aCSF (RaCSF; 300 mosmol kg$^{-1}$) and left at RT before transferring slices into the recording bath; RaCSF contained (in mM): 125 NaCl, 25 NaHCO$_3$, 10 D-glucose, 3 KCl, 1.2 KH$_2$PO$_4$, 1.2 MgSO$_4$ and 2 CaCl$_2$. Fire-polished thick-walled borosilicate glass pipettes (3–6 M$\Omega$; BF150-86-10; Sutter Instruments, Novato, CA, USA) were pulled using a P-1000 Flaming/Brown (Sutter Instruments) and filled with an internal solution, containing (in mM); 135 K-gluconate, 6 NaCl, 4 NaOH, 10 Hepes, 11 EGTA, 2 Mg-ATP, 0.3 Na$_2$-GTP, 1 MgCl$_2$ and 1 CaCl$_2$, pH 7.3, adjusted with KOH (293 mosmol kg$^{-1}$). Liquid junction potentials were corrected post recording ($-6.8$ mV at 32°C).

## Whole cell patch clamp electrophysiology

All recordings were made in open, whole cell configuration in voltage and current clamp mode. Signals were acquired using Multiclamp 700B amplifier and Axon Digidata 1440 (Axon Instruments, San Jose, CA, USA). All recordings occurred in carbogenated RaCSF at 32°C. Drugs were perfused (3 mL min$^{-1}$) over the slice in RaCSF. These included dopamine hydrochloride (10, 30 and 100 $\mu$M; H8502; Sigma, St Louis, MO, USA), capromorelin (10 nM; CP-424391; Pfizer, New York, NY, USA), serotonin hydrochloride (5-HT; 5 $\mu$M; H9523; Sigma) and $\alpha$-MS maleate salt ($\alpha$-MS; 5 $\mu$M; M110; Sigma). Resulting changes in current (pA) were recorded with $V_{Holding}$ at $-60$ mV. In experiments assessing post-synaptic effects of dopamine, tetrodotoxin citrate (TTX; 1 or 3 $\mu$M; 14964; Cayman Chemical, Ann Arbor, MI, USA) application occurred 5 min prior to dopamine. Neurons with an access resistance >25 M$\Omega$ were excluded, as were neurons with more than 50 pA leak current. Under voltage clamp conditions, access resistance change was monitored during the recording and neurons were excluded if access resistance changed >20%. Under current clamp conditions, bridge balance and pipette neutralisation compensation were performed.

## Data and statistical analyses

For HCR RNA-FISH and RNAscope assays, images were taken using a 20× objective on a Zeiss Axioscan7 slide scanner (Zeiss, Oberkochen, Germany) or using a 40× oil objective on a Zeiss Axioimager Z2 (Zeiss), respectively, and then quantified using Fiji (Fig. 1*A*-D). Neurons containing single or clustered dots were interpreted as positive for the target RNA and those without as negative, and overlaps between transcripts are represented as a percentage. Images for localisation experiments were taken using a 20× objective on a Zeiss Axioimager Z2 (Zeiss) (Fig. 2*A*-*C*), whereas synaptic input experiments were imaged using a 63× oil objective on a LSM880 confocal laser scanning microscope (Zeiss) (Fig. 2*E* and *F*). Quantification of boutons around D2R neurons in the lumbosacral defecation centre was carried out using Fiji. Cell borders were manually traced using the tdTomato signal to outline the cell body and dendrites. The proximity (distance in $\mu$m) of tyrosine hydroxylase (TH) and 5-HT boutons to the soma and primary dendrites of D2R positive neurons was examined. Bouton distances from the soma or primary (1°) dendrites were placed into bins from 0.2 $\mu$m to 2 $\mu$m. Colocalisation and

synaptic input data are represented as the mean ± SD and colocalisation data are presented as a percentage.

Electrophysiology data were collected and analysed with pClamp 10 and Clampfit (Molecular Devices, San Jose, CA, USA). Analysis of changes in holding currents were compared between baseline and peak drug response regions (mean pA) using Clampfit (Molecular Devices). Statistical analysis in Fig. 3*H* and *I*, Fig. 4*F–I* and Fig. 5*C* and *F* was performed on holding current changes in neurons at baseline (prior to drug wash on) and at peak of drug response and are presented as holding current changes relative to baseline [holding current Δ (pA)].

Analysis in Fig. 3*G* and Fig. 5*B* and *E* was performed on holding current changes in neurons at baseline (prior to drug wash on) and at peak of drug response. For these analyses, paired Student's *t* test were performed on holding currents at peak of response relative to baseline from the same cell. For many of the comparisons, we have analysed data from neurons in which the magnitudes of changes in response to agonists was > −5 pA, with these being considered responsive cells that had an inward excitatory current, with *P* values stated where appropriate. For comparisons of the effect across the entire population, *P* values are stated in the Results section. For Fig. 4*H*

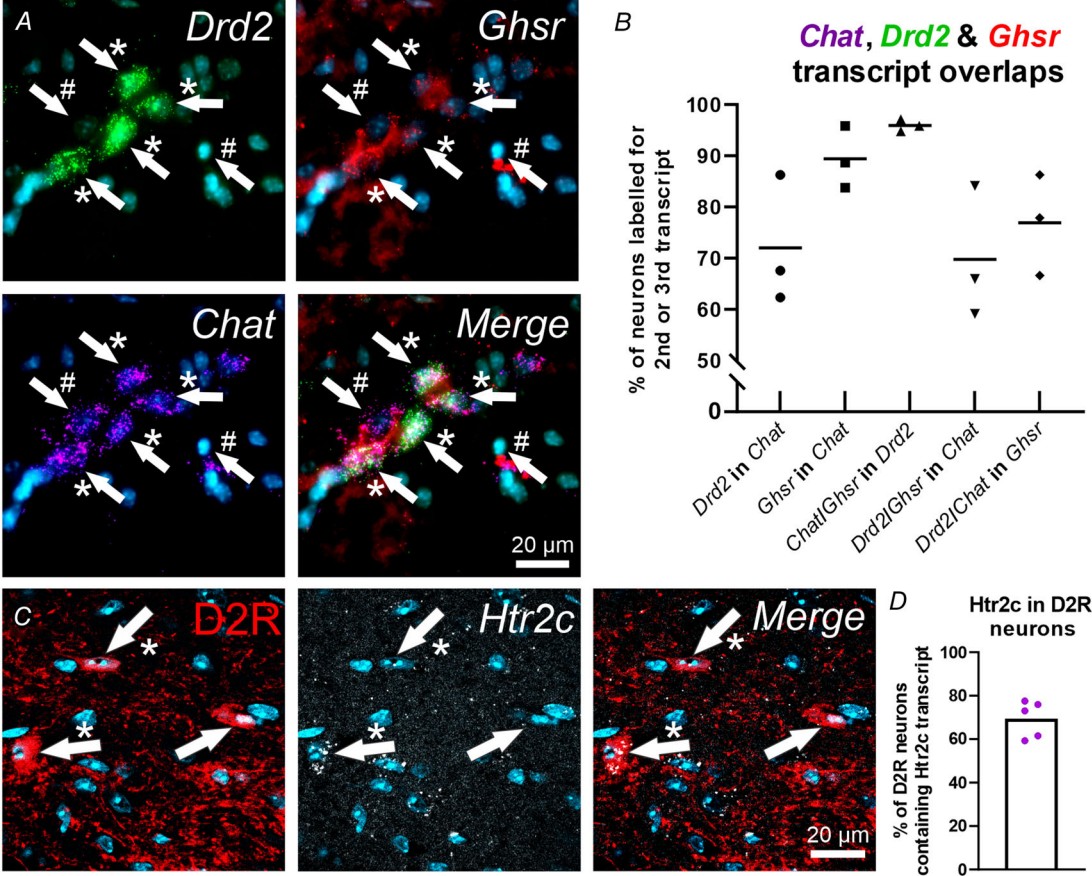

**Figure 1. PGNs in the adult mouse lumbosacral defecation centre region contain *Drd2*, *Ghsr* and *Ht2rc* mRNA transcript**

*A*, representative localisation of *Drd2*, *Ghsr* and *Chat* mRNA in the region of the lumbosacral defecation centre. Positive neurons are indicated by arrows. Arrows with a hash symbol (#) indicate neurons containing only *Ghsr* and *Chat*. Arrows with an asterisk (*) indicate neurons co-expressing *Drd2* (green), *Ghsr* (red) and *Chat* (magenta). DAPI staining in light blue. Images taken at 40× magnification. *B*, quantification of the proportional overlaps of transcript in parasympathetic PGNs in the region of the lumbosacral defecation centre (represented as a percentage of neurons labelled for second or third transcript). About 70% of PGNs, identified by *Chat*, also expressed *Drd2* and/or *Ghsr* in the lumbosacral defecation centre (*n* = 3 mice, 37 sections stained and analysed, 350 cells counted; mean ± SD). *C*, representative image of Htr2c expression in drd2-tdTomato (D2R) reporter mouse neurons of the lumbosacral defecation centre. Arrows indicate D2R positive neurons. Arrows with an astrisk (*) indicate D2R positive neurons (red) with *Ht2rc* transcript (white). Images taken at 20× magnification. *D*, quantification of Htr2c transcript in D2R neurons in the lumbosacral defecation centre. Around 70% of D2R neurons contained the transcript for *Htr2c* (*n* = 5 mice, 55 sections stained and analysed, 211 cells counted). Data represent the mean (SD values stated in text). [Colour figure can be viewed at wileyonlinelibrary.com]

and *I*, a paired Student's *t* test comparing the effect of dopamine with dopamine + YIL781 or dopamine repeat applications was performed on holding current changes relative to baseline and washout values. Exact *P* values are stated to three significant figures even when 'no statistical significance' is being reported. $P < 0.05$ was considered statistically significant. All data are represented as the mean ± SD and animal and cell numbers are either stated in the Results section or the figure legend. IPSCs and EPSCs were analysed using Mini analysis (Synaptosoft, Inc., Decatur, GA, USA). A paired Student's *t* test was performed on frequency change (Δ) during a matched time period during baseline and peak drug response (Fig. 3*J*). All statistical analysis were carried out using Prism, version 9 (GraphPad Software Inc., San Diego, CA, USA). Images of neurons in slices were taken using a 40× objective at the beginning of the recording, and diameter (μm) was measured at the two widest points using Fiji (https://fiji.sc).

## Results

### *Drd2*, *Ghsr* and *Ht2rc* transcripts are present in neurons in the lumbosacral defecation centre

We used RNAscope assays to investigate the expression of *Drd2*, *Ghsr* and *ChAT* mRNA transcripts in the lumbosacral defecation centre of adult mice (8 weeks old) (Fig. 1*A* and *B*). Parasympathetic PGNs were identified by the presence of choline *O*-acetyltransferase transcripts (*Chat*), an effective marker of preganglionic neurons in the spinal cord, all of which are cholinergic and express ChAT (Barber et al., 1984). *Drd2* was present in 72 ± 12% of PGNs (data presented as the mean ± SD), whereas *Ghsr* transcript was present in 89 ± 6% of PGNs. Some 95 ± 1% of PGNs that contained *Ghsr* also contained transcript for *Drd2* and 70 ± 13% of PGNs contained transcripts for both *Ghsr* and *Drd2*. Some 76 ± 10% of neurons containing *Ghsr* were positive for *Drd2* and *Chat* (n = 3 mice, 350 cells counted) (Fig. 1*B*).

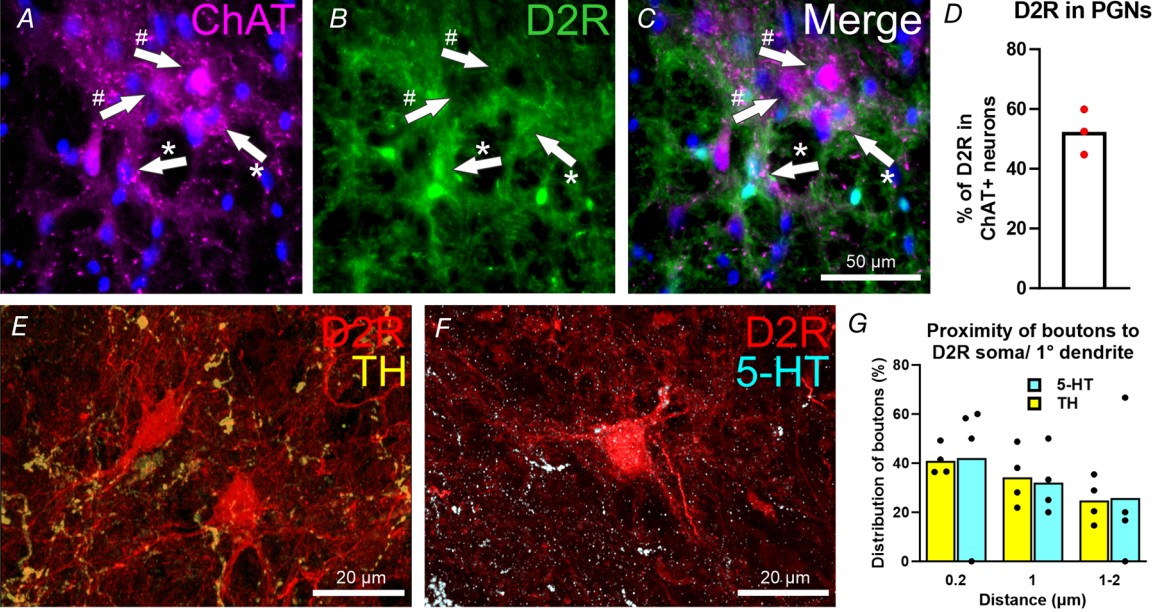

**Figure 2. D2R neurons in the adult mouse lumbosacral defecation centre are surrounded by TH and 5-HT boutons**

D2R expression in parasympathetic PGNs. Representative images showing co-occurrence of ChAT and D2R in cell bodies. *A*, ChAT-IR neurons labelled with Alexa Fluor 647 (magenta). *B*, D2R neurons expressing tdTomato (green) under the control of the *Drd2* promoter. *C*, merged image of ChAT-IR (magenta) and D2R neurons (green) in the PGNs in the mouse lumbosacral defecation centre. Positively labelled neurons are indicated by the arrows. Arrows with a hash symbol (#) indicate a neuron expressing only ChAT. Arrows with an asterisk (*) indicate a neuron expressing both ChAT and D2R. DAPI staining in blue. *D*, quantification of D2R in PGNs. Around 50% of PGNs contained D2R (n = 3 mice, 39 sections stained and analysed, 800 cells counted). *E*, TH boutons (yellow) in close apposition to a D2R positive (red) neuron in the mouse lumbosacral defecation centre. *F*, 5-HT boutons (cyan) in close apposition to D2R positive (red) neuron in the mouse lumbosacral defecation centre. *G*, quantification of the distribution of 5-HT (cyan) and TH (yellow) boutons within 2 μm of D2R neurons soma/1° dendrite (n = 4 mice, 76 sections stained and analysed, 70 5-HT-receiving cells, 573 TH-receiving cells analysed). Data represent the mean (SD values stated in text). [Colour figure can be viewed at wileyonlinelibrary.com]

**Table 1. Size and biophysical properties of tonic and phasic firing D2R neurons in the adult mouse lumbosacral defecation centre**

|  | Tonic firing neurons ($n = 54$) | Phasic firing neurons ($n = 14$) |
| --- | --- | --- |
| Diameter ($\mu$m) | $15.7 \pm 2.6$ | $15.9 \pm 3.4$ |
| Soma profile size ($\mu$m$^2$) | $124.8 \pm 38.0$ | $131 \pm 45.3$ |
| RMP (mV) | $-53.5 \pm 25.7$ ($-60.3^*$) | $-49.2 \pm 27.7$ ($-56^*$) |
| Input capacitance (pF) | $15.7 \pm 11.8$ | $18.4 \pm 7.5$ |

Data are presented as the mean $\pm$ SD.
Abbreviation: RMP, resting membrane potential.
$^*$RMP with corrected liquid junction potential.

HCR RNA-FISH was used to investigate *Ht2rc* transcript levels in drd2-tdTomato (D2R) reporter mice (12 weeks old) (Fig. 1*C*). Quantification revealed 69 $\pm$ 8% of D2R expressing neurons contained *Ht2rc* transcript ($n = 5$ mice, 211 cells counted) (Fig. 1*D*).

### D2R neurons in the mouse lumbosacral defecation centre are innervated by catecholamine and serotonergic boutons

Our next aim was to determine if D2R-expressing neurons in the adult mouse lumbosacral defecation centre were closely approached by boutons containing 5-HT and TH, a key, rate-limiting enzyme for catecholamine biosynthesis. First, immunolabelling in the lumbosacral defecation centre of drd2-tdTomato (D2R) reporter mice revealed 52 $\pm$ 8% of parasympathetic PGNs, identified by ChAT labelling, contained D2R ($n = 3$ mice, 800 cells counted) (Fig. 2*A–D*). This is comparable to our observations using RNAscope where 72 $\pm$ 12% of neurons containing *ChAT* transcripts expressed transcripts for *Drd2* (Fig. 1*B*). TH and 5-HT were observed within boutons in close apposition to D2R neurons in the lumbosacral defecation centre from single, stacked optical sections (Fig. 2*E* and *F*). Of the D2R neurons analysed, 70 $\pm$ 5% had TH boutons within 2 $\mu$m of a D2R soma/primary (1°) dendrite ($n = 4$ mice, 573 cells analysed) and 54 $\pm$ 10% had 5-HT boutons within 2 $\mu$m of a D2R soma/1° dendrite ($n = 4$ mice, 70 cells analysed). Further quantification revealed that of the TH boutons less than 2 $\mu$m from a D2R neuron soma or 1° dendrite, 41 $\pm$ 6% were at less than 0.2 $\mu$m, 34 $\pm$ 11% were at less than 1 $\mu$m and 25 $\pm$ 9% were between 1 and 2 $\mu$m (Fig. 2*G*). For 5-HT boutons less than 2 $\mu$m, 42 $\pm$ 28% were at less than 0.2 $\mu$m, 32 $\pm$ 13%, were at less than 1 $\mu$m and 26 $\pm$ 28% were between 1 and 2 $\mu$m. Given the extensive branching of PGNs in the lumbosacral spinal cord (Derjean et al., 2005), as well as the fact that not all branches could be identified and some were not included in the section, this is likely an under representation of the proportion of D2R neurons receiving catecholamine and/or serotonin innervation in the mouse lumbosacral defecation centre.

### Effects of dopamine and capromorelin in D2R neurons in the adult mouse lumbosacral defecation centre

D2R neurons in the adult mouse lumbosacral defecation centre were located using epifluorescence and Dodt contrast and recorded using whole cell electrophysiology (Fig. 3*A–C*). Biophysical properties of D2R neurons identified two populations (tonic and phasic firing neurons) based on their firing properties evoked by current injection steps (400 ms, $-80$ to 120 pA, 10 pA steps) (Table 1), as described previously in neonatal rats (Miura et al., 2000). There was no biologically relevant nor statistically significant ($P = 0.587$, unpaired $t$ test) difference in diameter and soma sizes between tonic and phasic firing PGNs. Resting membrane potential was similar between tonic ($-53.5 \pm 25.7$ mV) and phasic ($-49.2 \pm 27.7$ mV) firing PGNs ($P = 0.584$, unpaired $t$ test). Input capacitance was also similar ($P = 0.418$, unpaired $t$ test). Dopamine (30 $\mu$M) wash on under voltage clamp conditions ($V_{\text{Holding}}$ at $-60$ mV) caused an increase in the frequencies of IPSCs and EPSCs in a subpopulation of D2R neurons (Fig. 3*D*, *E* and *J*), but very little net effect on the holding current across the population (mean amplitude at peak of response relative to baseline ($1 \pm 13.7$ pA, $n = 18$ cells, five mice; $P = 0.003$, paired $t$ test $<-5$ pA, $P = 0.84$, paired $t$ test all cells) (Fig. 3*H*). EPSC frequencies increased by 2.4 $\pm$ 3.1 Hz relative to baseline ($P = 0.189$, paired $t$ test all cells) and IPSC frequencies increased by 4.5 $\pm$ 8.8 Hz ($P = 0.106$, paired $t$ test all cells) relative to baseline but across the entire population increases in frequencies were not statistically significant (Fig. 3*J*). The amplitudes of IPSCs and EPSCs were unchanged during dopamine wash on. When membrane potentials of neurons were adjusted to $-40$ mV ($V_{\text{Holding}}$), amplitudes of IPSCs increased as a result of the increased driving force for Cl$^-$. For $V_{\text{Holding}}$ at $-60$ mV, the driving potential for Cl$^-$ was calculated as $-1.59$ mV at 32°C. This indicates that, in some D2R neurons, there was presynaptic activity of dopamine.

In the following experiments, addition of TTX (1$-$3 $\mu$M) blocked any increases in the frequencies of IPSCs and EPSCs and revealed the slowly developing

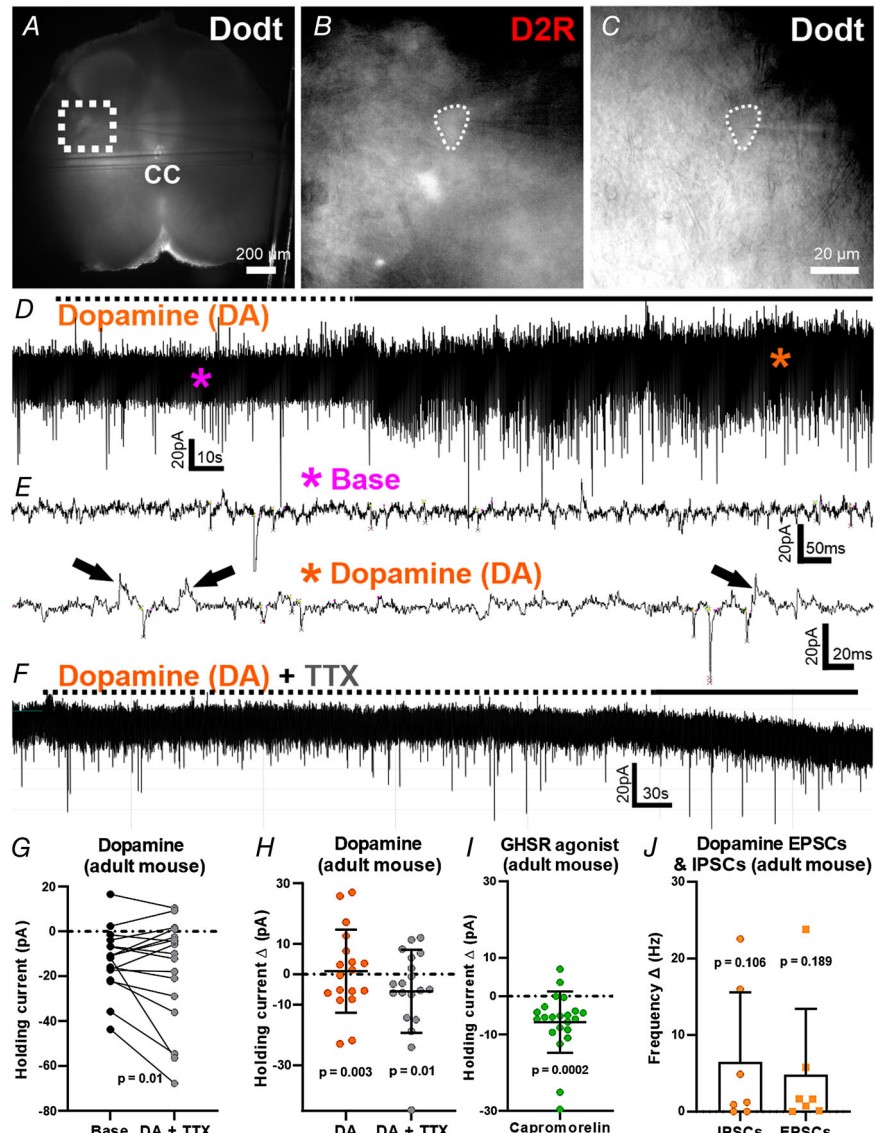

**Figure 3. Dopamine and capromorelin induce inward currents in D2R neurons in the adult mouse lumbosacral defecation centre**

*A*, representative Dodt contrast image at 5× magnification showing L6–S1 lumbosacral spinal cord region. White dashed box contains the lumbosacral defecation centre and recording region. *B*, 40× magnification of a recorded D2R neuron (white dashed line). *C*, 40× magnification image of the same neuron using Dodt contrast. *D*, representative trace of holding current changes in voltage clamp ($V_{Holding} = -60$ mV). Dopamine (DA) (30 $\mu$M) wash on induced increases in IPSCs and EPSCs in some D2R neurons. Dashed line indicates start of drug wash on, continuous line indicates drug effect period. *E*, parts of the trace in (*D*) on expanded timescales [indicated by an asterisk (*)] highlighting IPSCs (marked by arrows) and EPSCs at base and in the presence of dopamine (DA). *F*, 1 $\mu$M TTX pre-incubation reveals an inward change in holding current caused by dopamine (DA) (30 $\mu\mu$). Dashed line indicates start of drug wash on, continuous line indicates drug effect period. *G*, showing holding current changes (pA) between base and dopamine (DA) at peak of response in the presence of TTX (DA + TTX; *P* value represents paired *t* test on neurons with $<-5$ pA response. *H*, quantification of changes in holding current, $\Delta$ (pA), in response to dopamine under TTX (DA + TTX; 1–3 $\mu$M; *p* value represents paired *t* test on neurons with $<-5$ pA response) and non-TTX (DA; paired *t* test on neurons with $<-5$ pA response) conditions at peak of drug response compared to baseline. *I*, quantification of changes in holding current, $\Delta$ (pA), in response to capromorelin at peak of response compared to baseline (10 nM; *P* value represents paired *t* test on neurons with $<-5$ pA response). *J*, analysis of IPSCs and EPSCs frequency change ($\Delta$) in non-TTX conditions comparing peak of dopamine response to baseline (*n* = 7 cells, three mice, *p* value represents paired *t* test all cells). Data represent the mean $\pm$ SD. Abbreviations. CC, central canal; D2R, dopamine two receptor; TTX, tetrodotoxin; Dodt, Dodt contrast; GHSR, ghrelin receptor; IPSCs, inhibitory post synaptic currents; EPSCs, excitatory postsynaptic currents, base, baseline. [Colour figure can be viewed at wileyonlinelibrary.com]

**Table 2. Current responses of D2R neurons to dopamine under voltage clamp conditions in the presence and absence of TTX (1–3 $\mu$M)**

| Condition | Response, n (%) | pA change, relative to baseline (mean $\pm$ SD) |
|---|---|---|
| Dopamine | Inward, 8 (44%) | $-10.5 \pm 7.4$ |
| (18 cells, five mice) | Outward, 5 (28%) | $18 \pm 8.3$ |
| | No detectable change, 5 (28%) | $2.4 \pm 1.8$ |
| Dopamine in pres. of TTX | Inward, 10 (53%) | $-14.6 \pm 12.5$ |
| (19 cells, five mice) | Outward, 5 (27%) | $8.8 \pm 2.8$ |
| | No detectable change, 4 (21%) | $-1.3 \pm 2.4$ |

Inward current, $>-5$ pA, outward current, $>5$ pA, no change, $-5$ to 5 pA.

inward holding current in D2R neurons (mean amplitude at peak of response relative to baseline, $-5.6 \pm 13.7$ pA, $n = 19$ cells, five mice; $P = 0.01$, paired $t$ test $<-5$ pA, $P = 0.19$, paired $t$ test all cells) following dopamine wash on (Fig. 3*F*, *G* and *H* and Table 2). A heterogeneity of slow changes in holding current to dopamine (inward, outward, or no detectable change) was observed that was similar in TTX and non-TTX treated groups (Table 2). Except for dopamine, evoked outward holding currents ($18 \pm 8.3$ pA, mean amplitude at peak of response relative to baseline) were greater in amplitude in non-TTX treated D2R neurons compared to TTX treated D2R neurons ($8.8 \pm 2.8$ pA, mean amplitude at peak of response relative to baseline). There were also smaller dopamine induced inward change in holding currents in non-TTX treated D2R neurons (Table 2). Addition of the GHSR agonist, capromorelin (10 nM) induced an inward holding current in D2R neurons (mean amplitude at peak of response relative to baseline, $-7.4 \pm 8.1$ pA, $n = 22$ cells, six mice; $P = 0.001$, paired $t$ test $<-5$ pA and $P = 0.0007$, paired $t$ test all cells) (Fig. 3*I*). This GHSR evoked excitation occurred in the same D2R neurons exhibiting dopamine evoked inward holding current.

### Effects of dopamine and capromorelin in lumbosacral PGNs in neonatal (P7–14) rats

We investigated the effects of dopamine and the GHSR agonist, capromorelin, in retrogradely labelled PGNs in neonatal (P7–14) rats. PGNs in the lumbosacral defecation centre were back labelled from the pelvic organs using the lipophilic tracer, Fast DiI (Fig. 4*A* and *B*). PGNs were present on the lateral edge and/or medial part of Rexed laminae V and VII in the lumbosacral spinal cord (L6–S1) and were distinguishable from somatic motoneurons based on their sizes and locations. Biophysical properties of PGNs from neonatal (P7–14) rats were divided into two populations based on their firing properties (tonic or phasic) by current

injection steps, as described previously (Miura et al., 2000) (Table 3). There was a statistically significant difference ($P = 0.025$, unpaired $t$ test) between soma diameter when comparing tonic ($20 \pm 10.6 \mu$m) to phasic ($26.2 \pm 12.1 \mu$m) firing neonatal (P7–14) rat PGNs. Resting membrane potential was similar between tonic ($-54.3 \pm 19.4$ mV) and phasic ($-53.5 \pm 16.6$ mV) firing PGNs ($P = 0.886$, unpaired $t$ test). Input capacitance was also similar ($P = 0.195$, unpaired $t$ test). Dopamine wash on (30 $\mu$M) in current clamp mode ($V_{\text{Holding}}$ at $-60$ mV) caused a depolarisation and an increase in action potential firing frequency in the PGN that was washed off with RaCSF (Fig. 4*C*). Dopamine (10 $\mu$M) and capromorelin (10 nM) both induced inward holding currents in PGNs under voltage clamp conditions ($V_{\text{Holding}}$ at $-60$ mV) (Fig. 4*D* and *E*). Dopamine induced currents at the peak of response, relative to baseline, were quantified for 10 $\mu$M ($-4.6 \pm 21.5$ pA; 28 cells, seven rats; $P = 0.015$, paired $t$ test $<-5$ pA, $P = 0.267$, paired $t$ test all cells), 30 $\mu$M ($-5.8 \pm 10.8$ pA; 95 cells, 15 rats; $P \leq 0.0001$, paired $t$ test $<-5$ pA, $P \leq 0.0001$, paired $t$ test all cells) and 100 $\mu$M ($-2.5 \pm 13.9$ pA; 12 cells, three rats; $P = 0.037$, paired $t$ test $<-5$ pA, $P = 0.549$, paired $t$ test all cells) and pooled doses ($-5.3 \pm 13.8$ pA; 135 cells, 25 rats; $P \leq 0.0001$, paired $t$ test $<-5$ pA, $P \leq 0.0001$ paired $t$ test all cells) (paired $t$ test on holding currents at peak of response relative to baseline from the same cell) (Fig. 4*F*). Dopamine (30 $\mu$M) produced the largest inward current of the dopamine concentrations tested ($P \leq 0.0001$, paired $t$ test $<-5$ pA, $P \leq 0.0001$, paired $t$ test all cells) (Fig. 4*F*). Dopamine induced inward, excitatory holding currents in 53% of PGN, outward, inhibitory currents in 7% of PGNs, and 40% of PGNs showed no detectable change (Table 4). Capromorelin (10 nM) produced an inward holding current (mean amplitude at peak of response relative to baseline, $-2.9 \pm 7$ pA, 37 cells, 11 rats; $P \leq 0.0001$, paired $t$ test, $P \leq 0.0001$, paired $t$ test all cells) with no observable effects on EPSCs/IPSCs frequency (Fig. 4*E*). In the 63% of PGNs in which dopamine induced

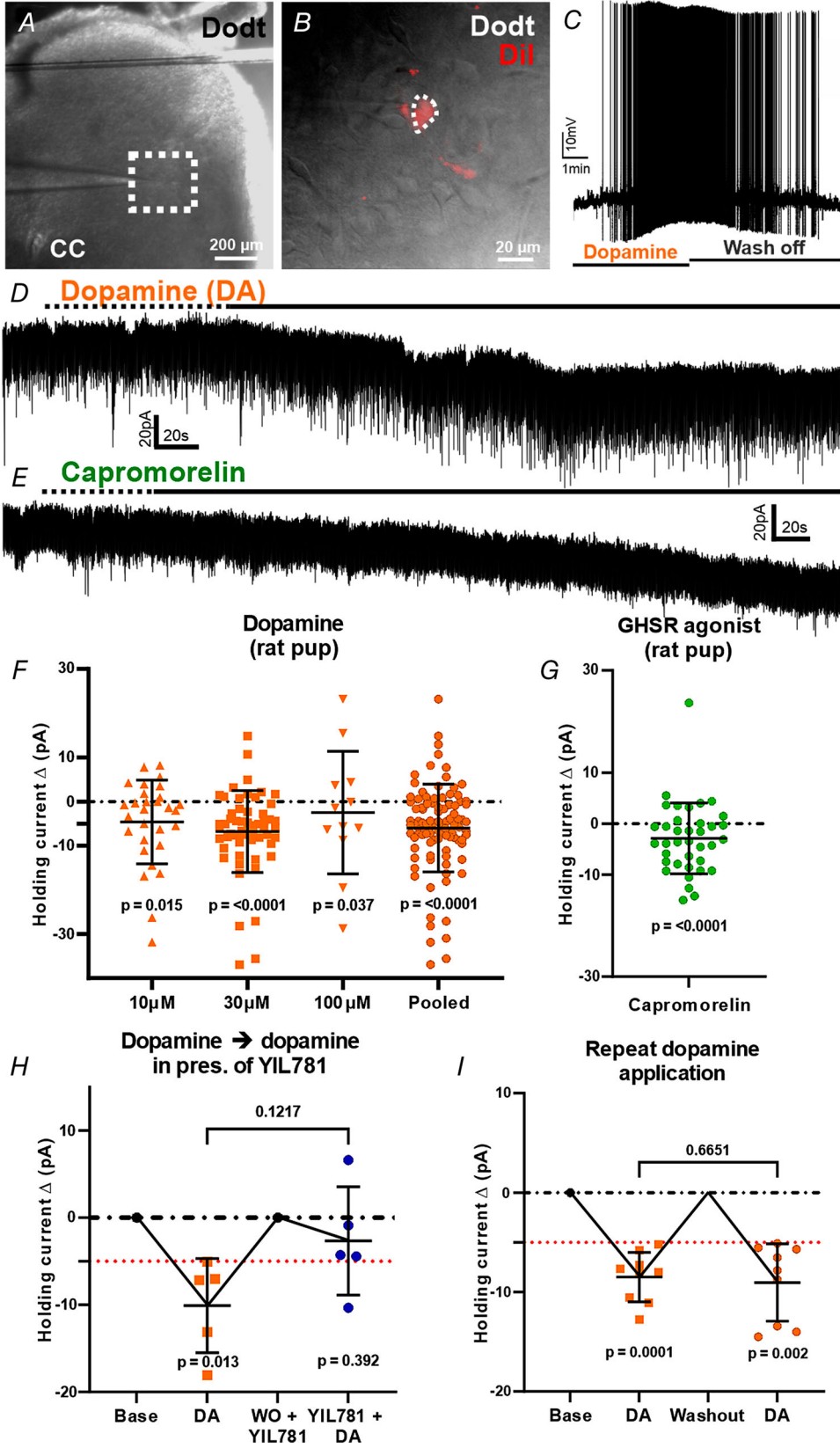

**Figure 4. Dopamine and capromorelin induce a postsynaptic inward holding current in identified PGNs of neonatal (P7–14) rats**

*A*, Dodt contrast image showing lumbosacral spinal cord (L6–S1). White dashed box highlights lumbosacral defecation centre. *B*, superimposed Dodt contrast image (white)/Fast DiI (red) back labelled PGN (white dashed line) recorded using whole cell electrophysiology. *C*, representative current clamp trace of dopamine (DA) (30 $\mu$M) producing sustained depolarisation of a PGN until drug was washed off with RaCSF. *D*, representative voltage clamp trace of dopamine (10 $\mu$M) wash on inducing an inward excitatory current. Dashed line indicates start of drug wash on, continuous line indicates drug effect period. *E*, representative voltage clamp trace of GHSR agonist (capromorelin; 10 nM) wash on inducing an inward current. Dashed line indicates start of drug wash on, continuous line indicates drug effect period. *F*, quantification of changes in holding current, $\Delta$ (pA), in response to dopamine at peak of response compared to baseline under increasing concentrations (10, 30 and 100 $\mu$M, and pooled doses) (pooled doses, *n* = 135 cells, 25 rats); *p* value represents paired *t* test on neurons with <−5 pA response. *G*, quantification of changes in holding current, $\Delta$ (pA), in response to capromorelin at peak of response compared to baseline (*n* = 15 cells, 11 rats; *p* value represents paired *t* test on neurons with <−5 pA response). *H*, holding current changes, $\Delta$ (pA) in PGNs where dopamine wash on induces an inward excitatory current, followed by dopamine washout (WO)/GHSR antagonist wash on (YIL781, 1 $\mu$M) and reapplication of dopamine (30 $\mu\mu$) and YIL781 (1 $\mu$M). GHSR antagonism blocked dopamine induced inward currents. Paired *t* test performed between normalised dopamine and YIL781 + dopamine current values at peak of response (*P* = 0.1217), paired *t* test of DA and YIL781 + DA performed on holding currents at peak of response compared to baseline. *I*, holding current changes, $\Delta$ (pA), in PGNs exposed to repeat dopamine applications (30 $\mu$M; *n* = nine cells, five rats). Paired *t* test performed between normalised dopamine and normalised dopamine current values (*P* = 0.6651), paired *t* test of DA (first application) and DA (second application) performed on holding currents at peak of response compared to baseline. Repeat dopamine wash on does not reduce magnitude of inward current response to agonist. Red dashed line placed at −5 pA is considered to be excitatory threshold and was used as a screen for subsequent YIL781 or dopamine application. Data represent the mean ± SD. Abbreviations: CC, central canal; Dodt, Dodt contrast; base, baseline; WO, washout. [Colour figure can be viewed at wileyonlinelibrary.com]

**Table 3. Biophysical properties of tonic and phasic firing PGNs in neonatal (P7–14) rats**

|  | Tonic firing neurons (*n* = 78) | Phasic firing neurons (*n* = 20) |
| --- | --- | --- |
| Diameter ($\mu$m) | 20.0 ± 10.6 | 26.2 ± 12.1 |
| Soma profile size ($\mu$m$^2$) | 208 ± 124 | 288 ± 139 |
| RMP (mV) | −54.3 ± 19.4 (−61.1*) | −53.5 ± 16.6 (60.3*) |
| Input capacitance (pF) | 27.0 ± 14.1 | 31.6 ± 13.9 |

Data are presented as the mean ± SD.
Abbreviation: RMP, resting membrane potential.
*RMP with corrected liquid junction potential.

**Table 4. Responses of PGNs in neonatal (P7–14) rats to dopamine under voltage clamp conditions**

| Condition | Response, *n* (%) | pA change, relative to baseline (mean ± SD) |
| --- | --- | --- |
| Dopamine (97 cells, 23 rats) | Inward, 51 (53%) | −12.0 ± 9.1 |
|  | Outward, 7 (7%) | 10.9 ± 6.2 |
|  | No detectable change, 39 (40%) | −1.1 ± 2.7 |

Inward current, >−5 pA, outward current, >5 pA, no change, −5 to 5 pA.

an inward shift in holding current, neurons were also excited by capromorelin. Furthermore, in PGNs where dopamine (30 $\mu$M) alone caused excitation, dopamine (30 $\mu$M; *P* = 0.013, paired *t* test all cells) in the presence of the GHSR antagonist, YIL781 (1 $\mu$M; *P* = 0.392, paired *t* test all cells), reduced inward excitatory currents (reduced to −2.7 ± 6.2 pA; five cells, three rats; *P* = 0.1217, paired *t* test performed between normalised dopamine

and YIL781 plus dopamine current values) (Fig. 4*H*). As a time and desensitisation control, we repeated applications of dopamine in dopamine responsive PGNs. The second application of dopamine did not reduce the magnitude of inward excitatory current produced (mean amplitude at peak of response relative to baseline, −8.4 ± 2.5 pA, first application; rats, *P* = 0.0001, paired *t* test all cells, −9.0 ± 3.9 pA, second application; nine cells, five rats,

$P = 0.002$, paired $t$ test all cells; $P = 0.6651$, paired $t$ test performed between normalised first and second application of dopamine) (Fig. 4*I*). These data support our observations *in vivo*, in which YIL781 blocked dopamine induced increases in colorectal motility (Furness et al., 2021).

### Effects of 5-HT and $\alpha$-MS in D2R neurons in the adult mouse lumbosacral defecation centre

Because both D2R and 5-HT agonist have been shown to be colokinetic *in vivo*, we investigated the effects of 5-HT and $\alpha$-MS in D2R neurons in the adult mouse lumbosacral defecation centre using whole cell electrophysiology. Under voltage clamp conditions ($V_{Holding}$ at $-60$ mV), 5-HT wash on induced mix responses, although, across the group on average, there were inward excitatory holding currents in D2R neurons (mean amplitude at peak of response relative to baseline, $-11.5 \pm 16.0$ pA; 14 cells, six mice; $P = 0.008$, paired $t$ test $<-5$ pA, $P = 0.116$, paired $t$ test all cells) (Fig. 5*A* and *B*). Similarly, in responses to $\alpha$-MS changes in holding current were variable (mean amplitude at peak of response relative to baseline, $-5.4 \pm 10.4$ pA; eight cells, two mice; $P = 0.024$, paired $t$ test $< -5$ pA, $P = 0.182$, paired $t$ test all cells) (Fig. 5*E*). A cohort of these D2R neurons exhibited inward holding currents with $\alpha$-MS (mean amplitude at peak of response relative to baseline, $-5.4 \pm 10.4$ pA; eight cells, two mice) (Fig. 5*E* and *D*). No detectable change in the frequency of IPSCs or EPSCs was observed following 5-HT or $\alpha$-MS wash on in D2R neurons. When tested on the same neuron, 5-HT and dopamine both induced inward holding currents (Fig. 5*C*) as did $\alpha$-MS and dopamine (Fig. 5*F*), suggesting that, at PGNs in the mouse lumbosacral defecation centre containing 5-HT2C and D2R, both mediate excitation.

### Discussion

Clusters of autonomic PGNs in the thoracic, lumbar and sacral spinal cord are surrounded by dopaminergic and serotoninergic terminals and many of these neurons express GHSR, which has been proposed to cause dopamine to be excitatory at D2R in neurons expressing both receptors (Furness et al., 2021). In the present study, we have used the lumbosacral spinal cord to investigate dopamine, serotonin and GHSR convergence at the same PGNs. Defecation control pathways pass through lumbosacral PGNs and are influenced by subcortical (pontomedullary) centres that receive cortical signals to maintain continence or to permit defecation, plus additional peripheral inputs; for example, from the stomach and large bowel (Browning & Travagli, 2014; Callaghan et al., 2018). Influences from the

pontomedullary centres are exerted on the contractile activity of the colorectum and via Onuf's nucleus on the striated muscle of the external anal sphincter. The nerve pathways from the pons and medulla that control colorectal propulsion relay through the defecation centre in the lumbosacral spinal cord (L6–S1), for which the output neurons are PGNs. Recent studies have identified the dopaminergic A11 cell group and the serotoninergic mid-line raphe cell group as controllers of PGNs in the defecation pathway (Nakamori et al., 2019; Nakamori, Naitou, Horii et al., 2018; Sawamura et al., 2023). In addition, ghrelin and GHSR agonists cause colorectal propulsion and defecation in animal models through actions at the level of the lumbosacral defecation centre (Charoenthongtrakul et al., 2009; Naitou et al., 2015; Pustovit et al., 2014; Shimizu et al., 2006) and are also colokinetic stimulants in humans (Acosta et al., 2016; Ellis et al., 2015). However, endogenous ghrelin is absent from the rodent spinal cord and other regions of the CNS (Cabral et al., 2017; Furness et al., 2011; Pustovit et al., 2017) and it has been suggested that a primary role of GHSR may be to modulate the effects of other transmitters, including dopamine, via *cis*-modulation between GPCRs (Kern et al., 2014). In rats, retrograde trans-synaptic labelling from the colon labelled two classes of neurons in the IML, identified as PGNs or inter-neurons (Vizzard et al., 2000). Thus, effects of dopamine, serotonin and GHSR agonists may be on separate PGNs that project to the colon, on the same PGNs, or on local interneurons in the lumbosacral defecation centre.

Stimulation of neurons in the dopaminergic A11 region of the brain stem causes increased contractile activity in the rat colon, an effect that is substantially inhibited by the 'D2R-prefering antagonist', haloperidol, injected intrathecally at the lumbosacral spinal level (Nakamori et al., 2019). Previous studies showed that PGNs identified by retrograde labelling from the colon were excited by dopamine and the D2R selective agonist, quinpirole (Naitou et al., 2016). In the present study, we confirmed that dopamine excites 53% PGNs, identified in the rat spinal cord by retrograde labelling, and found that dopamine, atypically, excites 44% of D2R-expressing neurons in the adult mouse IML. We have also shown that neurons expressing the D2R in this region are juxtaposed by boutons containing TH, a key enzyme for dopamine biosynthesis. Furthermore, intrathecal application of dopamine or quinpirole at the lumbosacral level causes colorectal propulsion and defecation (Furness et al., 2021; Naitou et al., 2016). In addition, work by Sawamura et al. (2023) showed that stimulation of A11 neurons expressing an inhibitory DREADD (hM4Di) reduces the number of faecal pellets produced in conscious animals, and also reduces colorectal propulsion caused by an intraluminal application of capsaicin to the colon. Together, these

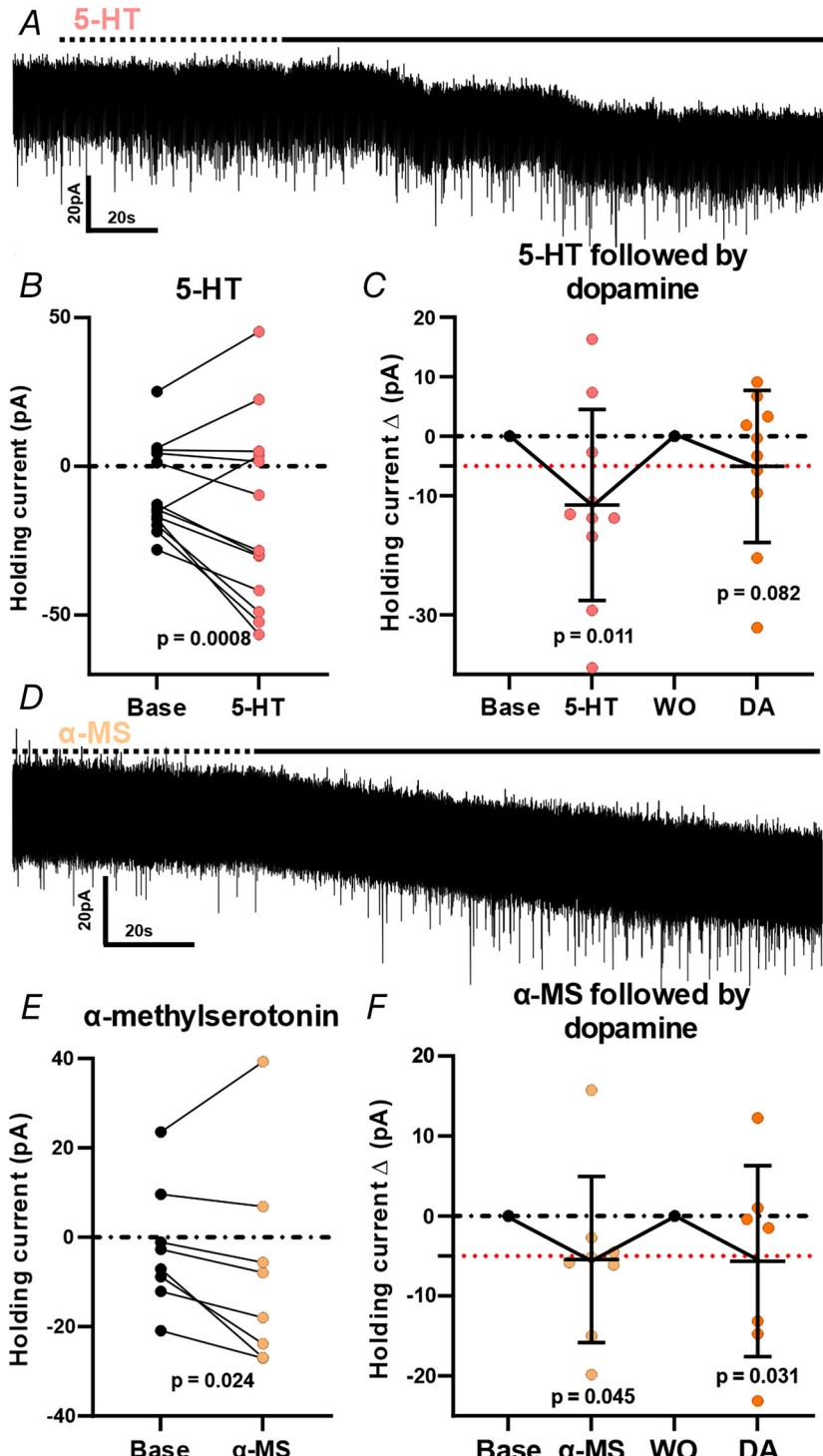

**Figure 5. 5-HT and 5HT2 agonist (α-MS) induce an inward holding current in a cohort of D2R neurons in the mouse lumbosacral defecation centre**

*A*, representative voltage clamp trace of 5-HT (5 $\mu$M) wash on inducing an inward excitatory current in a D2R neuron in the mouse lumbosacral defecation centre. Dashed line indicates start of drug wash on; continuous line indicates drug effect period. *B*, quantification of holding currents (pA) in baseline and 5-HT recordings (*n* = 14 cells, six mice; *p* value represents paired *t* test on neurons <−5 pA). *C*, quantification of changes in holding current, Δ (pA), following 5-HT (5 $\mu$M) and dopamine (DA) (30 $\mu$M) wash on in the same D2R neurons. *P* value represents paired *t* test on neurons with <−5 pA response. 5-HT and dopamine induce an inward holding current in a cohort of D2R neurons in the lumbosacral defecation centre. *D*, representative voltage clamp trace of α-MS (5 $\mu$M) wash

on inducing an inward excitatory current in a D2R neuron in the lumbosacral defecation centre. *E*, quantification of holding currents (pA) in baseline and $\alpha$-MS recordings (*n* = 8 cells, two mice; *p* value represents paired *t* test on neurons with $<-5$ pA response). *F*, quantification of changes in holding current $\Delta$ (pA) following $\alpha$-MS (5 $\mu$M) and dopamine (30 $\mu$M) wash on in the same D2R neurons. $\alpha$-MS and dopamine cause increased excitability in a cohort of D2R neurons in the lumbosacral defecation centre. *P* value represents paired *t* test on neurons with $<-5$ pA response. Red dashed line placed at $-5$ pA (considered as excitatory threshold). Data represent the mean $\pm$ SD. A paired student's *t* test was performed on holding currents at baseline compared to peak of current response following drug wash on. Abbreviations: $\alpha$-MS, $\alpha$-methylserotonin; 5-HTM, 5-hydroxytryptamine; base, baseline; WO, washout. [Colour figure can be viewed at wileyonlinelibrary.com]

data indicate that dopaminergic A11 neurons innervate PGNs that are in defecation stimulating pathways and excite the PGNs through D2R. We found that the same PGNs expressed GHSR. We have previously reported that the colokinetic effect of dopamine receptor agonists applied directly to the lumbosacral spinal level is blocked by systemic injection of the GHSR antagonist, YIL781 (Furness et al., 2021). The whole cell electrophysiology results of the present study show that dopamine induces an atypical inward excitatory current that can be blocked by YIL781 in PGNs of the lumbosacral defecation centre, suggesting that the presence of GHSR may be required for excitation at D2R, which is usually inhibitory in the CNS (Neve et al., 2004). It is possible that co-expressed GHSR may also interact with the 5-HT receptor, but we have not yet investigated this possibility in the lumbosacral spinal cord.

In addition to *Drd2* and *Ghsr*, we identified that PGNs at the level of the lumbosacral defecation centre contain mRNA transcripts for the serotonin receptor gene, *Htr2c*, and that serotonin receptor agonists (serotonin and $\alpha$-MS) induce an inward holding current in these neurons. This indicates that serotonergic, dopaminergic and GHSR-mediated effects converge on some of the same PGNs. Moreover, we observed that a majority of neurons expressing D2R in this region are juxtaposed by boutons containing serotonin. It has been reported previously that electrical stimulation of the midline raphe in the pontomedullary region, where serotonergic neurons that send projections to the lumbosacral defecation centre are located, causes colorectal contractions that are inhibited by serotonin receptor blockers, ketanserin and dolasetron (Nakamori, Naitou, Sano et al., 2018).

We observed that dopamine applied to D2R neurons in the adult mouse lumbosacral defecation centre induced large TTX-sensitive increases in the frequencies of IPSCs and EPSCs in some PGNs. This suggests that, in spinal cord slices from adult mouse, dopamine excites D2R-expressing PGNs directly and also those neurons that synapse to them. Gladwell & Coote (1999a,b) have described indirect excitatory and direct inhibitory effects of dopamine in sympathetic PGNs in thoracolumbar spinal cord slices of young rats. Intracellular recordings showed dopamine induced a slow depolarisation, which was reversed both by the 'D2R-prefering' antagonist

haloperidol, as well as under TTX conditions, suggesting that D2R-expressing interneurons drive excitation (Gladwell & Coote, 1999a,b). In D2R neurons in the lumbosacral defecation centre, we observed an opposite effect of dopamine in adult mice. We find that dopamine induces an inward holding current, as revealed in the presence of TTX, suggesting postsynaptic PGNs expressing D2R drive excitation. These observations suggest that indirect excitation through D2R occurs at thoracolumbar levels, and direct excitation occurs at lumbosacral levels.

We conclude that pontine dopaminergic and midline raphe serotoninergic neurons converge on some PGNs of the defecation pathways, where serotonin and dopamine are both excitatory. The effect of dopamine, exerted at D2R, is excitatory because of the presence of interacting ghrelin receptors. A modulatory role of GHSR might explain why its expression is conserved across vertebrates (mammals, birds, reptiles and fish) and is colokinetic in multiple species (dogs, humans and rats) (Kaiya et al., 2013; Sanger & Furness, 2016), supporting the concept for a physiological role of *cis*-modulation amongst GPCRs. Moreover, it may be feasible to develop therapies that target combinations of D2R, GHSR and serotonin receptors for the treatment of disorders of autonomic control.

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

# Additional information

### Data availability statement

All data supporting the results of the present study are provided in the manuscript and all data are available from the corresponding author upon reasonable request.

### Competing interests

The authors declare that they have no competing interests.

### Author contributions

J.B.F., S.G.B.F., S.J.M. and M.T.R. conceived of the study. Data were acquired by M.T.R. and A.K. M.T.R. and A.K. analysed data. MTR, S.J.M. and J.B.F. wrote the original manuscript. The manuscript was reviewed by all authors. M.T.R. generated figures. J.B.F. and S.G.B.F. provided funding. J.B.F. and S.J.M. provided resources and supervision. All authors approved the final version of the manuscript submitted for publication and agree to be accountable for all aspects of the work in ensuring that questions related to the accuracy or integrity of any part of the work are appropriately investigated and resolved. All persons designated as authors qualify for authorship, and all those who qualify for authorship are listed.

### Funding

This work was supported by 2021 NHMRC Ideas Grant; Furness 2021/GNT2012657.

### Acknowledgements

SGBF is an ARC Future Fellow (FT180100543).

Open access publishing facilitated by The University of Melbourne, as part of the Wiley - The University of Melbourne agreement via the Council of Australian University Librarians.

### Keywords

dopamine, ghrelin, lumbosacral spinal cord, serotonin, whole cell electrophysiology

### Supporting information

Additional supporting information can be found online in the Supporting Information section at the end of the HTML view of the article. Supporting information files available:

**Peer Review History**

