## [Peer Review History · The Journal of Physiology]

Sites and mechanisms of action of colokinetics at dopamine, ghrelin, and serotonin receptors in the rodent lumbosacral defecation centre

Mitchell Ringuet, Ada Koo, Sebastian GB Furness, Stuart J McDougall, and John B Furness

DOI: 10.1113/JP285217

Corresponding author(s): Mitchell Ringuet (mringuet@student.unimelb.edu.au)

The following individual(s) involved in review of this submission have agreed to reveal their identity: Kirsteen N Browning (Referee #1); Yasutake Shimizu (Referee #2)

Review Timeline:

Submission Date:	04-Jul-2023
Editorial Decision:	28-Jul-2023
Revision Received:	25-Aug-2023
Editorial Decision:	07-Sep-2023
Revision Received:	11-Sep-2023
Accepted:	13-Sep-2023

Senior Editor: *Peying Fong*

Reviewing Editor: *Bernard Drumm*

Transaction Report:

Dear Mr Ringuet,

Re: JP-RP-2023-285217 "Sites and mechanisms of action of colokinetics in the lumbosacral defecation centre" by Mitchell Ringuet, Ada Koo, Sebastian GB Furness, Stuart J McDougall, and John B Furness

Thank you for submitting your manuscript to The Journal of Physiology. It has been assessed by a Reviewing Editor and by 2 expert referees and we are pleased to tell you that it is acceptable for publication following satisfactory revision.

REVISION CHECKLIST:

We look forward to receiving your revised submission.

Yours sincerely,

Dr Peying Fong
Senior Editor
The Journal of Physiology
<https://jp.msubmit.net>
<http://jp.physoc.org>
The Physiological Society
Hodgkin Huxley House
30 Farringdon Lane
London, EC1R 3AW
UK
<http://www.physoc.org>
<http://journals.physoc.org>

REQUIRED ITEMS

-Author photo and profile. First (or joint first) authors are asked to provide a short biography (no more than 100 words for one author or 150 words in total for joint first authors) and a portrait photograph. These should be uploaded and clearly labelled with the revised version of the manuscript. See Information for Authors for further details.

-Please upload separate high-quality figure files via the submission form.

-Please include a legend to accompany your abstract figure.

-A Data Availability Statement is required for all papers reporting original data. This must be in the Additional Information section of the manuscript itself. It must have the paragraph heading "Data Availability Statement". All data supporting the results in the paper must be either: in the paper itself; uploaded as Supporting Information for Online Publication; or archived in an appropriate public repository. The statement needs to describe the availability or the absence of shared data. Authors must include in their Statement: a link to the repository they have used, or a statement that it is available as Supporting Information; reference the data in the appropriate sections(s) of their manuscript; and cite the data they have shared in the References section. Whenever possible the scripts and other artefacts used to generate the analyses presented in the paper should also be publicly archived. If sharing data compromises ethical standards or legal requirements then authors are not expected to share it, but must note this in their Statement. For more information, see our Statistics Policy.

EDITOR COMMENTS

Reviewing Editor:

This study reports novel and potentially clinically important insights into the expression of dopamine, serotonin and ghrelin receptors in preganglionic neurons of the lumbosacral spinal cord. When activated, these induce pathways that lead to the agonists act as 'colokinetics' to enhance colonic motility. Both reviewers commended the rigorous conduction of the experiments and highlighted the novel aspects of this understudied area of GI physiology. The reviewers have however highlighted some concerns which I agree should be addressed by the authors.

For example, I agree with the suggestion of reviewer 2 to amend the title of the paper to be more specific for the data obtained (at the very least state the species to which the data relate to).

The biggest concern, raised by both reviewers were the relatively low numbers of animals used in some of the electrophysiological experiments, which have made it difficult to interpret the statistical significance in many comparisons. While the authors state that they record currents for example from multiple neurons in each experiment (up to a dozen or more), in some datasets this might be from as few as 2-3 animals. This becomes an issue when in some data sets, such as in Fig 3 the authors claim a lack of effect of interventions but close examination of the data suggest that in some instances that might be accounted for by variability in a small biological n number and thus more replicates would be required to truly make comparisons before reaching a final conclusion. I would be in favour of suggesting an increase in biological n values to have a minimum of 4-5 animals per dataset for the electrophysiological experiments. This would remove the concern of reviewer 1 in particular that results should not be reported as having no effect if statistical significance is not reached with a small sample size.

Both reviewers also specifically highlight the data contained in Fig. 3 to be confusing and in need to be reworked and overall the statistical tests used, definition of n and the specific details of experimental protocols need to be more explicitly stated in the Methods. Additionally some of the language used to describe the electrophysiological phenomenon mentioned by reviewer 1 (inward current vs inward change in holding potential) should be reconsidered carefully as this point may be confusing to readers.

Senior Editor:

Two Expert Referees and a Reviewing Editor provide their detailed assessments of your manuscript. They agree on its potential impact to drive further inquiry in what presently is an important, yet understudied, arena. Also noteworthy is the rigor applied to data acquisition.

Nonetheless, concerns were raised that require attention. These fall within two general categories:

1) the low number of biological replicates (animal numbers) was noted. There is some room for improvement in this regard (RE and Referee 1);

2) phrasing and presentation (including presentation of statistical methods) can be clarified; such concerns likely can be addressed readily by taking detailed Referee comments into account and judicious editing (RE, and both Referees).

With regard to comments pertaining to statistical tests employed: I do see that references to the tests are embedded within some of the Figure Legends. It may be useful to direct readers to this information within a separate section in the Methods.

Thank you for submitting your study to The Journal of Physiology. We look forward to receiving your revised manuscript.

REFEREE COMMENTS

Referee #1:

Conceptually, its not explicitly clear why some experiments were performed in neonatal rats rather than performing all experiments in mice. It is recognized that the previous literature showing common sites of action of amines and ghrelin were performed in rats, but these experiments were essentially replicated in the current manuscript in mice and it would appear the recording from backlabeled lumbosacral PGNs could equally be performed in mice rather than neonatal rats.

Immunohistochemistry and RNA Scope - please provide details of how many sections per mouse were stained and analyzed.

Analysis and statistic methodology. Please note at the earliest point that results are expressed as mean (SD), and also note at what value statistical significance was set (assumed $P < 0.05$?). Please also note the statistical test used when analyzing the immunohistochemistry/RNA-FISH/RNA-Scope results, and include this information in the figure/table legends as required.

Figure 3 (and onward) - I understand the need to include all data points on the graphical summaries (eg 3G-I) but the symbols used are incredibly confusing and it is not immediately obvious what the differences are between the closed circles, squares (eg Figure 3G; open circle, closed circle but with a black border $P = 0.19$, close circle no black border $P = 0.01$; the

same applies to 3H in terms of closed squares, one of which seems smaller than the other? Does the p value associated with both the open and closed symbol refer to the combined data and that associated with the closed symbol refers to the inward current data only?). I would suggest using entirely different shapes to represent outward current vs no current vs inward current which may help in distinguishing the different recordings.

Figure 3J is also confusing in terms of previously stated results "The transient events were identified as inhibitory postsynaptic currents (IPSCs) whose frequencies increased by 2.4 (3.1) Hz relative to baseline, and excitatory postsynaptic currents, (EPSCs), whose frequencies increased by 4.5 (8.8) Hz relative to baseline (Fig. 3E, H, I)". Were these changes significant (no stats reported)? My assumption is that Figure 3J is showing changes in EPSCs/IPSC frequency following dopamine superfusion but appears to imply these changes were not actually significant, in which case, Figure 3D appears to be an outlier and not a representative trace. Alternatively, there are two groups of cells; ones in which dopamine increases EPSCS and IPSCS (were these always the same cells as suggested by Figure 3D) and ones in which dopamine had no effect. The relatively small number of cells in the effects of dopamine were examined on synaptic currents may preclude such population segregation, however. It also appears that Figure 3J includes data from 7 cells, not 6 as the legend states.

Page 19: "Dopamine wash on (30uM) in current clamp mode (Vholding at -60mV) caused a depolarization of the PGN which was washed off with RaCSF (Figure 1C)" This data is not shown (or not shown in Figure 1C).

Page 19 "Dopamine induced currents at the peak of response, relative to baseline, were quantified by 10uM.....30uM.....100uM.... and pooled doses. Dopamine (30uM) produced the largest inward current of the dopamine concentrations tested" In how many neurons were multiple concentrations of dopamine tested? And which statistical test was used to compare these results (an ANOVA with post hoc tests would seem appropriate but these details are not included in the methodology or in the figure legends)

Page 19 The manuscript would benefit from a very clear distinction being made between inward currents (EPSC) and inward change in holding potential. Non-neurophysiologists may have difficulty in distinguishing the two and this is particularly problematic when noting that a drug (dopamine or capromorelin) induces "inward excitatory currents" when it has previously been stated that dopamine may also increase EPSC frequency. The authors note "Capromorelin (10nM) produced inward excitatory currents.....-2.9 (7) pA" Given the previous classification of a responding neuron as having an effect >5pA, it would appear that capromorelin was not effective in all neurons. It may be important to note how many neurons responded to capromorelin and dividing the results into responding vs non-responding neurons. The authors also note "In PGNs where dopamine (30 μM) induced inward excitatory currents, subsequent capromorelin application (10 nM) also induced an inward excitatory current. 63% of PGNs that produced inward excitatory currents to dopamine were also excited by capromorelin" These two statements are somewhat contradictory but may be better rephrased as "In the 63% of PGNs in which dopamine induced an inward shift in holding current, neurons were also excited by capromorelin"

Page 19 "Furthermore, in PGNs where dopamine....alone caused excitation, dopamine in the presence of the GHSR antagonist.....caused a reduce inward excitatory current (reduced -2.7 (6.2)pA" Was the current reduced by -2.7 (6.2)pA or reduced to -2/7 (6.2)pA? Please add in the statistical details (as included in Figure 4 and legend).

Referee #2:

This study show that dopamine, serotonin, and ghrelin receptors are expressed in the same PGNs in the lumbosacral spinal cord. In addition, the authors demonstrate that dopamine, serotonin, and ghrelin receptor agonists activate overlapping populations of lumbosacral PGNs. This study has shed important new light on the regulatory mechanism of defecation reflex. The results are novel and important. I only have several minor comments.

1. I think the title is too broad. I would like the authors to consider making the title appropriate for the content of this study.

2. The authors showed that dopamine-induced excitation was reversed by GHSR antagonism. This may be one of the most important findings in this study. I would like to know whether the GHSR antagonism also inhibit serotonin-induced excitation or not. If the authors already have data, please include them in this paper. If there is no data available at this time, I would very much hope that the authors would address this issue in the future. I am not necessarily requesting additional experiments for this paper.

3. Since ilium xylazil (P.6) is the trade name, please correct it to "xylazine hydrochloride".

4. The description "snap frozen" (P.8, L.5) is not enough to give details, so please add something more specific, such as on dry ice or in liquid nitrogen.

5. To clarify the technique used, please include the words "patch clamp" in the method related to whole cell electrophysiology.

6. In Fig2 A-C, magenta and red are used for ChAT and D2R, respectively, but these two colors are indistinguishable (especially Marge is a problem). Please change them to Magenta vs Cyan or Red vs Green.

7. In Fig. 2G, there is a description of (μ M), but this may be a mistake for (μ m)? In some parts of the results, μ m is also written as μ M. Please check and correct them.

8. I think the result's description of Fig. 3 (P. 15) dose not correspond exactly to the figure. There seems to be some error, so please check carefully and revise the manuscript.

9. Explanation for Fig. 4C is missing in the RESULT section. Please add it.

END OF COMMENTS

Confidential Review

04-Jul-2023

Responses to Editors and Reviewers' comments

Manuscript No: JP-RP-2023-285217

Original Title: Sites and mechanisms of action of colokinetics in the lumbosacral defecation centre

By: Mitchell Ringuet, Ada Koo, Sebastian GB Furness, Stuart J McDougall, and John B Furness

We have copied all the reports, including comments from the Senior and Reviewing Editors below and have responded to each point (red, italic).

Reviewing Editor

This study reports novel and potentially clinically important insights into the expression of dopamine, serotonin and ghrelin receptors in preganglionic neurons of the lumbosacral spinal cord. When activated, these induce pathways that lead to the agonists act as 'colokinetics' to enhance colonic motility. Both reviewers commended the rigorous conduction of the experiments and highlighted the novel aspects of this understudied area of GI physiology. The reviewers have however highlighted some concerns which I agree should be addressed by the authors.

For example, I agree with the suggestion of reviewer 2 to amend the title of the paper to be more specific for the data obtained (at the very least state the species to which the data relate to).

Response: We are pleased to receive the positive comments. We have amended the title of the paper, as requested, to "Sites and mechanisms of action of colokinetics at dopamine, ghrelin, and serotonin receptors in the rodent lumbosacral defecation centre".

The biggest concern, raised by both reviewers were the relatively low numbers of animals used in some of the electrophysiological experiments, which have made it difficult to interpret the statistical significance in many comparisons. While the authors state that they record currents for example from multiple neurons in each experiment (up to a dozen or more), in some datasets this might be from as few as 2-3 animals. This becomes an issue when in some data sets, such as in Fig 3 the authors claim a lack of effect of interventions but close examination of the data suggest that in some instances that might be accounted for by variability in a small biological n number and thus more replicates would be required to truly make comparisons before reaching a final conclusion. I would be in favour of suggesting an increase in biological n values to have a minimum of 4-5 animals per dataset for the electrophysiological experiments. This would remove the concern of reviewer 1 in particular that results should not be reported as having no effect if statistical significance is not reached with a small sample size.

Response: It is true that some of the data is from low numbers of animals, and we agree with the RE that results should not be reported as having 'no effect' if statistical significance is not reached with a small sample size. We have removed statements of no effect where few animals were used. Small groups of experiments are reported where they guided the research. For example, we report that in a small number of experiments dopamine elicited large numbers of EPSCs and IPSCs. This was the observation that resulted in us conducting experiments in the presence of tetrodotoxin to investigate the post-synaptic actions of dopamine in isolation.

Both reviewers also specifically highlight the data contained in Fig. 3 to be confusing and in need to reworked and overall the statistical tests used, definition of n and the specific details of experimental protocols need to be more explicitly stated in the Methods. Additionally some of the language used to describe the electrophysiological phenomenon mentioned by reviewer 1 (inward current vs inward change in holding potential) should be reconsidered carefully as this point may be confusing to readers.

Response (see also specific response to Referee #2): We agree that the manuscript would benefit from a clearer distinction between inward transient currents, i.e. EPSCs and inward changes in holding current. We have changed the nomenclature from outward/ inward transient currents to the more direct terminology IPSCs and EPSCs. We have replaced "induced inward excitatory

currents” with “induced an inward change in the holding current” (or simply “induced an inward holding current” where that suited the context) to make it clear what was observed.

Senior Editor:

Two Expert Referees and a Reviewing Editor provide their detailed assessments of your manuscript. They agree on its potential impact to drive further inquiry in what presently is an important, yet understudied, arena. Also noteworthy is the rigor applied to data acquisition.

Nonetheless, concerns were raised that require attention. These fall within two general categories:

1) the low number of biological replicates (animal numbers) was noted. There is some room for improvement in this regard (RE and Referee 1);

Response: We have responded to the Reviewing Editor about the cases in which there were low numbers of biological replicates. We agree with the overall RE comments on replicates and have moderated our arguments accordingly

2) phrasing and presentation (including presentation of statistical methods) can be clarified; such concerns likely can be addressed readily by taking detailed Referee comments into account and judicious editing (RE, and both Referees).

Response: We agree with the comments and have responded below to the referee comments.

With regard to comments pertaining to statistical tests employed: I do see that references to the tests are embedded within some of the Figure Legends. It may be useful to direct readers to this information within a separate section in the Methods.

Response: We agree with the comments relating to statistical tests employed and have placed details within the figure legends in the method sections.

Referee #1:

Conceptually, it's not explicitly clear why some experiments were performed in neonatal rats rather than performing all experiments in mice. It is recognized that the previous literature showing common sites of action of amines and ghrelin were performed in rats, but these experiments were essentially replicated in the current manuscript in mice and it would appear the recording from backlabeled lumbosacral PGNs could equally be performed in mice rather than neonatal rats.

Response: The principal reason why we have included the rat studies is that previous in vivo studies that suggest physiologically important roles of ghrelin, dopamine and serotonin, including the dependence on the ghrelin receptor for excitation through the DRD2, have been conducted in rats. The relevant in vivo experiments that have been previously published have not been conducted in mice. Most experiments were conducted in mouse spinal cord slices because of the availability of a reporter mouse for DRD2. Back-labelling in young rats allowed us to investigate post-synaptic effects in anatomically identified PGNs and to ensure consistency in outcomes across species. Performing the same surgeries on immature mice was not regarded as feasible.

Immunohistochemistry and RNA Scope - please provide details of how many sections per mouse were stained and analyzed.

Response: In the revision, we have included details of numbers of mice and numbers of sections stained and analysed within each legend.

Analysis and statistic methodology. Please note at the earliest point that results are expressed as mean (SD), and also note at what value statistical significance was set (assumed $P < 0.05$?). Please also note the statistical test used when analyzing the immunohistochemistry/RNA-FISH/RNA-Scope results and include this information in the figure/table legends as required.

Response: We have added/ edited the analysis and statistics methodology section. We have added

the way in which statistics are expressed at the earliest point this is used: "Drd2 was present in 72 (12) % of PGNs (data presented as mean (SD)), while Ghnr transcript was present in 89 (6) %." In the methods we have also noted the P value threshold "...and statistical significance was set at $p < 0.05$ ". We have included information about analysis of RNAscope/ IHC in both the methods and the figure legends.

Figure 3 (and onward) - I understand the need to include all data points on the graphical summaries (eg 3G-I) but the symbols used are incredibly confusing and it is not immediately obvious what the differences are between the closed circles, squares (eg Figure 3G; open circle, closed circle but with a black border $P = 0.19$, close circle no black border $P = 0.01$; the same applies to 3H in terms of closed squares, one of which seems smaller than the other? Does the p value associated with both the open and closed symbol refer to the combined data and that associated with the closed symbol refers to the inward current data only?). I would suggest using entirely different shapes to represent outward current vs no current vs inward current which may help in distinguishing the different recordings.

We agree that the symbols used were difficult to interpret and the way that the p values were presented was not ideal. We have now simplified the graphs from Figure 3 onward by reporting the statistics of the combined data (inward ($< -5pA$) and outward and no change in holding current) in the main text and have changed all symbols to closed on the graphs. Additionally, we have reported the p value relating to the inward current only on the graphs to simplify the presentation for the reader.

Figure 3J is also confusing in terms of previously stated results "The transient events were identified as inhibitory postsynaptic currents (IPSCs) whose frequencies increased by 2.4 (3.1) Hz relative to baseline, and excitatory postsynaptic currents, (EPSCs), whose frequencies increased by 4.5 (8.8) Hz relative to baseline (Fig. 3E, H, I)". Were these changes significant (no stats reported)? My assumption is that Figure 3J is showing changes in EPSCs/IPSC frequency following dopamine superfusion but appears to imply these changes were not actually significant, in which case, Figure 3D appears to be an outlier and not a representative trace. Alternatively, there are two groups of cells; ones in which dopamine increases EPSCS and IPSCS (were these always the same cells as suggested by Figure 3D) and ones in which dopamine had no effect. The relatively small number of cells in the effects of dopamine were examined on synaptic currents may preclude such population segregation, however. It also appears that Figure 3J includes data from 7 cells, not 6 as the legend states.

Response: For Figure 3J we agree that these results could be confusing to the reader. These changes did not reach statistical significance due to the variability/ heterogeneity of changes in IPSCs/EPSCs following dopamine wash on. This is now indicated in the revised text, where more detail on the heterogeneity is now included. The observation of highly variable, sometimes large, increases in numbers of PSCs is important to document as this led to the use of tetrodotoxin to isolate the post-synaptic effects of dopamine. In line with the referee comment, we have changed the legend and stated 7 not 6 cells.

Page 19: "Dopamine wash on (30uM) in current clamp mode (V_{holding} at -60mV) caused a depolarization of the PGN which was washed off with RaCSF (Figure 1C)" This data is not shown (or not shown in Figure 1C).

Response: This was a typographical error. This results relates to Figure 4C, not Figure 1C. This error has been corrected, and we have added "...caused a depolarisation and an increase in action potential firing frequency"

Page 19 "Dopamine induced currents at the peak of response, relative to baseline, were quantified by 10uM.....30uM.....100uM.... and pooled doses. Dopamine (30uM) produced the largest inward current of the dopamine concentrations tested" In how many neurons were multiple concentrations

of dopamine tested? And which statistical test was used to compare these results (an ANOVA with post hoc tests would seem appropriate but these details are not included in the methodology or in the figure legends)

Response: Multiple dopamine exposures were applied in a total of 15 of the 135 neurons, 9 of which were used to test for potential tachyphylaxis of the dopamine response (Fig. 4I). We didn't compare across groups but instead compared each peak response for the given dose relative to the baseline holding current within the same cell. This has been included in the methodology and in the Fig. 4 legend " concentrations (10 μ M, 30 μ M, 100 μ M and pooled doses) (pooled data, n= 135 cells, 25 rats; paired t-test on holding currents at peak of response relative to baseline within the same cell)."

Page 19 The manuscript would benefit from a very clear distinction being made between inward currents (EPSC) and inward change in holding potential. Non-neurophysiologists may have difficulty in distinguishing the two and this is particularly problematic when noting that a drug (dopamine or capromorelin) induces "inward excitatory currents" when it has previously been stated that dopamine may also increase EPSC frequency. The authors note "Capromorelin (10nM) produced inward excitatory currents.....-2.9 (7) pA" Given the previous classification of a responding neuron as having an effect >5pA, it would appear that capromorelin was not effective in all neurons. It may be important to note how many neurons responded to capromorelin and dividing the results into responding vs non-responding neurons. The authors also note "In PGNs where dopamine (30 μ M) induced inward excitatory currents, subsequent capromorelin application (10 nM) also induced an inward excitatory current. 63% of PGNs that produced inward excitatory currents to dopamine were also excited by capromorelin" These two statements are somewhat contradictory but may be better rephrased as "In the 63% of PGNs in which dopamine induced an inward shift in holding current, neurons were also excited by capromorelin"

Response: We agree that the manuscript would benefit from a clearer distinction between inward transient currents, i.e. EPSCs, and inward holding current. We have changed the nomenclature from "induced inward excitatory currents" to "induced an inward change in holding current". We also removed the use of transient currents and defined them as EPSCs and IPSCs to help provide clear distinction for the reader. We have decided to remove the classification of "having an excitatory effect" from these graphs due to heterogeneity of responses across the population of neurons and instead report the p value relating to neurons that showed >-5 pA change following drug application. The referee was also correct about the ambiguity concerning neurons responding to both dopamine and capromorelin and we have changed the sentence to " In the 63% of PGNs in which dopamine induced an inward shift in holding current, neurons were also excited by capromorelin"

Page 19 "Furthermore, in PGNs where dopamine....alone caused excitation, dopamine in the presence of the GHSR antagonist.....caused a reduce inward excitatory current (reduced -2.7 (6.2)pA" Was the current reduced by -2.7 (6.2)pA or reduced to -2/7 (6.2)pA? Please add in the statistical details (as included in Figure 4 and legend).

Response: This was reduced to -2.7 (6.2) pA and has been corrected in the text. We have added the statistical test performed at the end of the sentence "(reduced to -2.7 (6.2) pA; 5 cells, 3 rats; paired t-test performed between normalised dopamine and YIL781 plus dopamine current values)".

Referee #2:

This study show that dopamine, serotonin, and ghrelin receptors are expressed in the same PGNs in the lumbosacral spinal cord. In addition, the authors demonstrate that dopamine, serotonin, and ghrelin receptor agonists activate overlapping populations of lumbosacral PGNs. This study has shed important new light on the regulatory mechanism of defecation reflex. The results are novel and important. I only have several minor comments.

1. I think the title is too broad. I would like the authors to consider making the title appropriate for the content of this study.

Response: We have changed the title, as suggested.

2. The authors showed that dopamine-induced excitation was reversed by GHSR antagonism. This may be one of the most important findings in this study. I would like to know whether the GHSR antagonism also inhibit serotonin-induced excitation or not. If the authors already have data, please include them in this paper. If there is no data available at this time, I would very much hope that the authors would address this issue in the future. I am not necessarily requesting additional experiments for this paper.

Response: We also consider this to be an important result. We investigated the GHSR antagonism because we were intrigued that excitation was mediated through the D2R, which is the opposite to the expected effect. Serotonin had its expected excitatory effect and we have not yet investigated whether responsiveness to serotonin could be modified by GHSR antagonism. We have commenced studies to investigate GHSR interactions with other G-protein coupled receptors, including in other CNS regions. We have added a comment about possible GHSR interaction with the serotonin receptor in the revised discussion.

3. Since ilium xylazil (P.6) is the trade name, please correct it to "xylazine hydrochloride".

Response: This has been corrected to the drug name in the immunohistochemistry and HCR-RNA FISH section of the methods.

4. The description "snap frozen" (P.8, L.5) is not enough to give details, so please add something more specific, such as on dry ice or in liquid nitrogen.

Response: More details have been added on (P7.L7) "Tissue was frozen in OCT using isopentane that was cooled with liquid nitrogen and coronal sections " and (P8.L5) "and snap frozen in OCT using isopentane that was cooled with liquid nitrogen"

5. To clarify the technique used, please include the words "patch clamp" in the method related to whole cell electrophysiology.

Response: Title in the method section has been changed from "whole cell electrophysiology" to "Whole cell patch clamp electrophysiology"

6. In Fig2 A-C, magenta and red are used for ChAT and D2R, respectively, but these two colors are indistinguishable (especially Merge is a problem). Please change them to Magenta vs Cyan or Red vs Green.

Response: We have edited Fig. 2A-C so that ChAT is magenta and D2R is green.

We have also changed Fig.1C so that Htr2c is white instead of magenta to help the reader distinguish between D2R and Htr2c.

7. In Fig. 2G, there is a description of (μ M), but this may be a mistake for (μ m)? In some parts of the results, μ m is also written as μ M. Please check and correct them.

Response: Thanks for picking this up. Fig. 2G has been changed from μ M to μ m (P13, L15) "5-HT boutons less than 2μ M, 42" changed to "5-HT boutons less than 2μ m, 42"

8. I think the result's description of Fig. 3 (P. 15) does not correspond exactly to the figure. There seems to be some error, so please check carefully and revise the manuscript.

Response: The referee is correct; we had made some errors in our referring to the Figure panels. These errors have been corrected.

9. Explanation for Fig. 4C is missing in the RESULT section. Please add it.

Response: We made an error and referred to the Fig. 4C as Fig. 1A. We have amended this error. We have also elaborated on the result from Fig. 4C. “Dopamine wash on (30 μ M) in current clamp mode (VHolding at -60 mV) caused a depolarisation and an increase in action potential frequency in the PGN that was washed off with RaCSF (Fig. 4C).”

Dear Mr Ringuet,

Re: JP-RP-2023-285217R1 "Sites and mechanisms of action of colokinetics at dopamine, ghrelin, and serotonin receptors in the rodent lumbosacral defecation centre" by Mitchell Ringuet, Ada Koo, Sebastian GB Furness, Stuart J McDougall, and John B Furness

Thank you for submitting your manuscript to The Journal of Physiology. It has been assessed by a Reviewing Editor and by 2 expert referees and we are pleased to tell you that it is acceptable for publication following satisfactory revision.

REVISION CHECKLIST:

Please upload two versions of your manuscript text: one with all relevant changes highlighted and one clean version with no changes tracked. The manuscript file should include all tables and figure legends, but each figure/graph should be uploaded as separate, high-resolution files. The journal is now integrated with Wiley's Image Checking service. For further details, see: <https://www.wiley.com/en-us/network/publishing/research-publishing/trending-stories/upholding-image-integrity-wileys-image-screening-service>.

We look forward to receiving your revised submission.

Yours sincerely,

Dr Peying Fong
Senior Editor
The Journal of Physiology
<https://jp.msubmit.net>
<http://jp.physoc.org>
The Physiological Society
Hodgkin Huxley House
30 Farringdon Lane
London, EC1R 3AW
UK
<http://www.physoc.org>
<http://journals.physoc.org>

REQUIRED ITEMS

Please provide a legend to accompany your abstract figure. You can include this in your article (Word) file.

EDITOR COMMENTS

Reviewing Editor:

The authors have addressed the comments of both reviewers, who have noted the potential impact of the study on this understudied area of gut physiology.

A figure legend is required for the abstract figure.

Senior Editor:

Thank you for your thorough responses to suggestions arising from review of the initial version of your manuscript. Both Referees are satisfied with your treatment of their feedback in this revised version, and overall feel the study offers impactful mechanistic insights into a long-understudied physiological process. Regarding my suggesting pertaining to presentation of statistics, the additional information provided within the Methods section is appreciated.

There is one remaining point, raised by the Reviewing Editor, and that regards the need for a legend to the Abstract Figure. Please do provide this in your final revision.

REFEREE COMMENTS

Referee #1:

The authors have answered all questions raised in the initial review and amended the manuscript accordingly. In particular, the revisions to the figures/symbols increases clarity significantly, and the distinction between transient inwards currents/inward shift in holding potential will be helpful for non-electrophysiologists.

I have no further questions/comments.

Referee #2:

Thank you for your appropriate response to my comments.

I appreciate that it is a very valuable paper.

END OF COMMENTS

1st Confidential Review

25-Aug-2023

Responses to Editors and Reviewers' comments

Manuscript No: JP-RP-2023-285217R1

Revised Title: Sites and mechanisms of action of colokinetics at dopamine, ghrelin, and serotonin receptors in the rodent lumbosacral defecation centre

By: Mitchell Ringuet, Ada Koo, Sebastian GB Furness, Stuart J McDougall, and John B Furness

We have copied all the reports, including comments from the Senior and Reviewing Editors below and have responded to each point (red, italic).

REQUIRED ITEMS

Please provide a legend to accompany your abstract figure. You can include this in your article (Word) file.

We have included a legend for the abstract figure on P3 of the manuscript JP-RP-2023-285217R1

Reviewing Editor:

The authors have addressed the comments of both reviewers, who have noted the potential impact of the study on this understudied area of gut physiology.

A figure legend is required for the abstract figure.

See above

Senior Editor:

Thank you for your thorough responses to suggestions arising from review of the initial version of your manuscript. Both Referees are satisfied with your treatment of their feedback in this revised version, and overall feel the study offers impactful mechanistic insights into a long-understudied physiological process. Regarding my suggesting pertaining to presentation of statistics, the additional information provided within the Methods section is appreciated.

There is one remaining point, raised by the Reviewing Editor, and that regards the need for a legend to the Abstract Figure. Please do provide this in your final revision.

See above

REFEREE COMMENTS

Referee #1:

The authors have answered all questions raised in the initial review and amended the manuscript accordingly. In particular, the revisions to the figures/symbols increases clarity significantly, and the distinction between transient inwards currents/inward shift in holding potential will be helpful for non-electrophysiologists.

I have no further questions/comments.

Referee #2:

Thank you for your appropriate response to my comments.

I appreciate that it is a very valuable paper.

Dear Dr Ringuet,

Re: JP-RP-2023-285217R2 "Sites and mechanisms of action of colokinetics at dopamine, ghrelin, and serotonin receptors in the rodent lumbosacral defecation centre" by Mitchell Ringuet, Ada Koo, Sebastian GB Furness, Stuart J McDougall, and John B Furness

We are pleased to tell you that your paper has been accepted for publication in The Journal of Physiology.

Authors should note that it is too late at this point to offer corrections prior to proofing. The accepted version will be published online, ahead of the copy edited and typeset version being made available. Major corrections at proof stage, such as changes to figures, will be referred to the Editors for approval before they can be incorporated. Only minor changes, such as to style and consistency, should be made at proof stage. Changes that need to be made after proof stage will usually require a formal correction notice.

Yours sincerely,

Dr Peiyong Fong
Senior Editor
The Journal of Physiology
<https://jp.msubmit.net>
<http://jp.physoc.org>
The Physiological Society
Hodgkin Huxley House
30 Farringdon Lane
London, EC1R 3AW
UK
<http://www.physoc.org>
<http://journals.physoc.org>

P.S. - You can help your research get the attention it deserves! Check out Wiley's free Promotion Guide for best-practice recommendations for promoting your work at www.wileyauthors.com/eeo/guide. You can learn more about Wiley Editing Services which offers professional video, design, and writing services to create shareable video abstracts, infographics, conference posters, lay summaries, and research news stories for your research at www.wileyauthors.com/eeo/promotion.

IMPORTANT NOTICE ABOUT OPEN ACCESS: To assist authors whose funding agencies mandate public access to published research findings sooner than 12 months after publication, The Journal of Physiology allows authors to pay an Open Access (OA) fee to have their papers made freely available immediately on publication.

You can check if your funder or institution has a Wiley Open Access Account here: <https://authorservices.wiley.com/author-resources/Journal-Authors/licensing-and-open-access/open-access/author-compliance-tool.html>.

EDITOR COMMENTS

Reviewing Editor:

Thank you for your final revisions and for submitting this work. All referee and RE comments have been addressed.

Senior Editor:

The Abstract Figure now has been complemented by a succinct legend, effectively addressing the remaining point raised by the Reviewing Editor. All concerns raised in review by both Expert Referees, as well as the Reviewing Editor, have been addressed, and your manuscript is ready for final acceptance.

We are pleased that you chose to submit your important study to The Journal of Physiology. Thank you.

2nd Confidential Review

11-Sep-2023